

# Quantitative estimate of sources of uncertainty in drone-based methane emission measurements

Tannaz H. Mohammadloo[1], Matthew Jones[1], Bas van de Kerkhof[1], Kyle Dawson[2], Brendan James Smith[2], Stephen Conley[3], Abigail Corbett[4], and Rutger IJzermans[1]

[1]Shell Global Solutions International B.V, Amsterdam, The Netherlands
[2]SeekOps Inc., Austin, Texas, United States of America
[3]Scientific Aviation, A Division of ChampionX, Boulder, Colorado, United States of America
[4]GTI Energy, Chicago, Illinois, United States of America

**Correspondence:** Rutger IJzermans (Rutger.Ijzermans@shell.com)

**Abstract.** Site level measurements of methane emissions are used by operators for reconciliation with bottom-up emission inventories with the aim to improve accuracy, thoroughness and confidence in reported emissions. In that context it is of critical importance to avoid measurement errors, and to understand the measurement uncertainty. Remotely piloted aircraft systems (commonly referred to as 'drones') can play a pivotal role in the quantification of site-level methane emissions. Typical implementations use the 'mass balance method' to quantify emissions, with a high-precision methane sensor mounted on a quadcopter drone flying in a vertical curtain pattern; the total mass emission rate can then be computed post hoc from the measured methane concentration data and simultaneous wind data. Controlled release tests have shown that errors with the mass balance method can be considerable. For example, Liu et al. (2023) report absolute errors for more than 100% for the two drone solutions tested; on the other hand, errors can be much smaller, of the order of 16% root-mean-square errors in Corbett & Smith (2022), if additional constraints are placed on the data, restricting the analysis to cases where the wind field was steady. In this paper we present a systematic error analysis of physical phenomena affecting the error in the mass balance method for parameters related to the acquisition of methane concentration data and to postprocessing. The sources of error are analysed individually, and it must be realised that individual errors can accumulate in practice, and they can also be augmented by other sources that are not included in the present work. Examples of these sources include the uncertainty in methane concentration measurements by a sensor with finite precision or the method used to measure the unperturbed wind velocity at the position of the drone. We find that the most important source of error considered is the horizontal and vertical spacings in the data acquisition as a coarse spacings can results in missing a methane plume. The potential error can be as high as 100% in situations where the wind speed is steady and the methane plume has a coherent shape, contradicting the intuition of some operators in the industry. The likelihood of the extent of this error can be expressed in terms of a dimensionless number defined by the spatial resolution of the methane concentration measurements and the downwind distance from the main emission sources. The learnings from our theoretical error analysis are then applied to a number of historical measurements in a controlled release setting. We show how the learnings on the main sources of error can be used to eliminate potential errors during the



postprocessing of flight data. Second, we evaluate an aggregated data set of 1,001 historical drone flights; our analysis shows that the potential errors in the mass balance method can be of the order of 100% on occasions, even though the individual errors can be much smaller in the vast majority of the flights. The discussion section provides some guidelines to industry on how to avoid or minimize potential errors in drone measurements for methane emission quantification.

## 1   Introduction

Methane is a much more potent greenhouse gas than carbon dioxide when comparing global warming potential (GWP). Methane has a GWP of 86 over the first 20 years since its atmospheric injection, and a GWP of 28 over a 100-year time frame (IPCC , 2014). Since the lifetime of methane in the atmosphere is just over 12 years (relatively short compared to $CO_2$'s lifetime of hundreds of years), reducing methane emissions now offers great potential for

delivering substantial reductions in global warming on a timescale compatible with the 2015 Paris Agreement goals. The average background level of methane in the atmosphere is about 1.9 ppm globally, increasing at about 0.01 ppm/year due to human activity (Nisbet et al. , 2019). About 60% of total methane emissions are anthropogenic, predominantly from agriculture and waste; oil and gas production accounts for 13% of total methane emissions (Saunois et al. , 2020).

Best practices for reporting of methane emissions are proposed in the OGMP2.0 reporting framework (United Nations Environment Programme , 2020); a voluntary, comprehensive, measurement-based reporting framework for the oil and gas industry. At the highest reporting level OGMP2.0 recommends building a measurement-based source-level inventory of emissions, perform an independent site-level emission measurement – and reconciliation of the two. The aim of reconciliation is to help improve accuracy in reported emissions as well as identifying emission reduction

opportunities. In this context it is important for operators to understand sources of error and uncertainty ranges in measurement techniques used. Sources of measurement errors should be avoided where possible and practical, and remaining uncertainties should be understood and correctly estimated. Site-level quantification is typically done with airborne methane measurements (e.g., on remotely piloted aircraft systems or on manned aircraft). Liu et al. (2023) note that the errors in quantification can be considerable: although many systems can quantify "within an

order of magnitude of the controlled release rate", the absolute errors reported varied between 19% and 239%. The two remotely piloted aircraft systems (drones) tested even reported results with errors of 140% and 239%. Corbett & Smith (2022) present the results from another controlled release campaign: they report errors in the methane emission rate up to 115%, but they also show that the errors can be much reduced, to about 16%, by considering only measurements during the time that the wind conditions were favorable. Still, Corbett & Smith (2022) do not

give a systematic analysis into the causes why some measurements are more successful than others.

The present paper provides a framework that allows quantitative assessment of errors in methane emission rate measurements using the mass balance method. We present a theoretical framework that provides an explanation for



the various sources of uncertainty considered, and highlights which of these effects have a particularly large impact on uncertainty – and thus are those effects that should be managed most carefully when setting up a drone survey campaign. The focus will be on the uncertainty related to the flight path and the variability of the wind; we do not take into account additional uncertainty associated with limited precision of the methane concentration sensor and the method used to determine the wind vector at the position of the drone (including anemometer precision).

The second part of this paper illustrates the various sources of uncertainty with real-life examples. Data from a controlled release experiment acquired by Scientific Aviation are analyzed to illustrate potential uncertainties and subtleties associated with the data analysis process. In addition, a data set of 1,001 historic flights acquired by SeekOps is analyzed with regards to the variations in the wind speed and direction variations during the survey time, and the potential effect of the choices for the drone's flight path on the quality of the results.

This paper intends to provide guidelines to prevent avoidable errors in data collection and data analysis of drone-based measurements of methane emissions from industrial facilities. Although a number of obvious sources of potential error have been included in our analysis, it must be recognized that other errors may also occur in practice. This means that the actual uncertainty in measurement results will always depend on the particular deployment of a solution at a given site on a certain day. Examples of these additional sources of error are a reduced performance of the sensor(s) due to weather conditions or external damage, and complications associated with turbulent air flow around buildings and pieces of equipment, and the method used to measure the unperturbed wind velocity at the position of the drone.

## 2 Methods

### 2.1 Coordinate system definition

Before describing the methods that are used for quantification of methane emissions, we first introduce the coordinate systems that will be used in the remainder of this paper.

The first coordinate system is the Local Geodetic Coordinate System ($XYZ$), with $X$-axis pointing to the East direction and $Y$-axis pointing to the North direction. $Z$-axis is defined such that a right-handed coordinates system is obtained.

The second coordinate system is aligned with the wind direction and referred to as ($\xi \eta z$). This means that the horizontal direction of the wind field is aligned with the $\xi$-axis, the direction perpendicular to it is given by $\eta$-axis. The $z$-axis is defined such that a right-handed system is obtained. The coordinate system is defined such that the emission source is located at $\xi = 0$, $\eta = 0$ and $z = H$ with $H$ being the height of the source, see Figure C2 in the Supplementary Information.

The third coordinate system is defined to represent the drone measurements and is a 2-dimensional coordinate system that captures the trajectory flown by the drone. In case of a 'curtain' flight path (a vertical plane), each coordinate on the curtain is expressed in $Pz$ coordinate system with $P$-axis parallel to the ground and $z$-axis the





vertical direction. In case of cylindrical flight paths, polar coordinates $\theta z$ are used with $z$ the vertical direction and $\theta$ the angular position in radians which can be converted to distance using the radius of the cylinder.

### 2.2 Mass balance method for methane emission quantification

The mass conservation equation for the methane mass concentration $c$ (mass of methane per volume air) is (Stockie , 2011):

$$\frac{\partial c}{\partial t} + \boldsymbol{\nabla} \cdot (c\mathbf{u}) = \boldsymbol{\nabla} \cdot (K\boldsymbol{\nabla}c) + \sum_i^{N_{\text{sources}}} \dot{m}_i \delta(\mathbf{x} - \mathbf{x}_{s,i}) \tag{1}$$

where $\mathbf{u}$ is the wind field vector and $K$ is the molecular diffusivity of scalar $c$. The mass emission rate of point source $i$ located at $\mathbf{x}_{s,i} = [x_{s,i}, y_{s,i}, z_{s,i}]$ is denoted by $\dot{m}_i$; $\delta$ denotes the Dirac delta function. The first term on the left-hand side denotes the change of concentration in time, whereas the second term denotes the change due to advection; the first term on the right hand side indicates molecular diffusion and the second term accounts for the presence of point sources.

The methane mass concentration $c$ is related to the molar volume of methane in air, $c'$ (with unit [ppmv]), through:

$$c = \frac{c'}{10^6} \times \frac{M_{\text{CH4}}}{M_{\text{air}}} \times \rho_{\text{air}} \tag{2}$$

where $M_{\text{CH4}}$ is the molar mass of methane ($16.04\,\text{kg/kmol}$) and $M_{\text{air}}$ is the molar mass of dry air (typically about $28.95\,\text{kg/kmol}$). The mass density of air, $\rho_{\text{air}}$, will change with differences in ambient temperature and pressure (i.e., $\rho_{\text{air}}(T, P)$).

We can obtain the mass balance equation for a volume by integrating:

$$\int_V \frac{\partial c}{\partial t} dV + \int_V \boldsymbol{\nabla} \cdot (c\mathbf{u}) dV = \int_V \boldsymbol{\nabla} \cdot (K\boldsymbol{\nabla}c) dV + \int_V \sum_i^{N_{\text{sources}}} \dot{m}_i \delta(\mathbf{x} - \mathbf{x}_{s,i}) dV \tag{3}$$

where the integrals on both sides have units of [kg/s]. The total methane emission from all sources inside the volume combined can be estimated by applying the divergence theorem (Katz , 1979):

$$\dot{m} = \sum_i^{N_{\text{sources}}} \dot{m}_i = \int_V \frac{\partial c}{\partial t} dV + \oint_{\partial V} c\mathbf{u} \cdot \mathbf{n} dS - \oint_{\partial V} K\mathbf{n} \cdot \boldsymbol{\nabla}c dS \tag{4}$$

Usually, the situation is considered where the main contribution on the right-hand-side comes from the advection term (i.e., the wind vector is aligned with the normal vector $\mathbf{n}$ at the surface where the concentration flows out, and $|c\mathbf{u} \cdot \mathbf{n}| \gg |K\mathbf{n} \cdot \boldsymbol{\nabla}c|$, so that the diffusion term can be neglected (Conley et al. , 2017). When the process is assumed



to be statistically stationary (i.e. the statistical properties of the process do not change with time, $\frac{\partial c}{\partial t}$), it is possible

to derive the mass balance equation (Conley et al. , 2017) for the total methane emission rate:

$$\dot{m} \approx \oint_{\partial V} c\mathbf{u} \cdot \mathbf{n} dS = 10^{-6} \frac{M_{\mathrm{CH4}}}{M_{\mathrm{air}}} \oint_{\partial V} c' \rho_{\mathrm{air}} \mathbf{u} \cdot \mathbf{n} dS, \tag{5}$$

where the latter part is obtained by substituting Equation (2).

We observe some inconsistencies in the presentation of the mass balance calculation in existing literature, for

instance with regards to the molar mass used (Corbett & Smith , 2022) or the distinction between concentrations

by weight and by volume (Negron et al , 2020). We hope to avoid further ambiguities by the full derivation of the

mass balance method, leading up to Equation (5).

### 2.3  Drone-based methane measurements

The relationship in Equation (5) does not require that dispersion happens in a particular way (e.g., a Gaussian

plume), nor that there is only one source within the volume. To apply this form of the mass concentration equation

to real problems, a flight path needs to be defined that approximates the boundary integral. In a common scenario,

the measurements are taken downwind of an equipment area under the assumption that everything entering the

upstream volume (behind the source) is atmospheric background, $c_0$. In some deployments, a cylindrical flight

pattern is followed that circumscribes the facility of interest (Corbett & Smith , 2022).

The terms inside the integral of Equation (5) must generally be approximated, owing to the discrete nature of

real concentration and wind measurements. We assume that the bounding surface (e.g., a cylinder or box around a

region of interest, or a plane downwind of a point of interest) can be partitioned into a two-dimensional regular grid

in the horizontal and vertical directions with spacings equal $\delta P$ and $\delta z$ respectively. The exact details of this grid

will depend on the flight trajectory under consideration. Approximating Equation (5) directly gives:

$$\dot{m} \approx \sum_{k=1}^{n_z} \sum_{j=1}^{n_P} (c_{j,k} - c_0)[\mathbf{u}_{j,k} \cdot \mathbf{n}_{j,k}]\delta P \delta z \tag{6}$$

where $c_{j,k}$ and $\mathbf{u}_{j,k}$ denote the methane mass concentration and the wind field interpolated onto the grid at location

$\mathbf{x}_{j,k}$ (either measured directly or interpolated onto a grid) and $\mathbf{u}_{j,k}$ denotes the wind field at the same location. $\mathbf{n}_{j,k}$

is the normal vector at the surface at location $\mathbf{x}_{j,k}$. $n_z$ and $n_p$ are the number of cells in the $P$ and $z$ directions

respectively. For molar concentration measurements, the calculation should be modified to take account of the density

of methane, as in Equation (5)

$$\dot{m} \approx 10^{-6} \frac{M_{\mathrm{CH4}}}{M_{\mathrm{air}}} \sum_{k=1}^{n_z} \sum_{j=1}^{n_P} (c'_{j,k} - c'_0)\rho_{\mathrm{air,j,k}}[\mathbf{u}_{j,k} \cdot \mathbf{n}_{j,k}]\delta P \, \delta z \tag{7}$$

where $\rho_{\mathrm{air,j,k}}$ represents the air density at the location $\mathbf{x}_{j,k}$, which may be estimated using the local temperature

and pressure.



Equation (6) and Equation (7) describe the calculation procedure in the mass balance method for one single flight
spanning the vertical plane of interest. If a similar trajectory along the vertical plane is flown multiple times, then
the mass emission rate can be obtained by averaging. The exact choice of averaging procedure is important for the
final result. Two common choices are:

1. $\dot{m}_\ell$ for each curtain $\ell$ separately using Equation (7), and then take the average of these values:

$$\dot{m}_{av,1} = \frac{1}{N_{\text{curtain}}} \sum_{\ell=1}^{N_{\text{curtain}}} \dot{m}_\ell \tag{8}$$

2. First compute the horizontal flux per altitude, $\phi_k$, i.e., $\phi_k = 10^{-6} \frac{M_{\text{CH4}}}{M_{\text{air}}} \sum_{j=1}^{n_y} (c'_{j,k} - c'_0) \rho_{\text{air,j,k}} [\mathbf{u}_{j,k} \cdot \mathbf{n}_{j,k}] \delta P \, \delta z.$
If multiple horizontal fluxes are available per an altitude, the median is taken $\tilde{\phi}_k = \text{median}(\phi_k)$. With median
horizontal flux, the emission rate is calculated as

$$\dot{m} = \sum_{k=1}^{n_z} \tilde{\phi}_k \delta z \tag{9}$$

The two methods are equivalent when the exact same trajectory is flown for each curtain. If there are gaps in the
data in the vertical direction, however, the two methods may result in different calculated methane emission rates.

Usually, a drone flies at a speed between $1\,\text{m/s}$ to $5\,\text{m/s}$ to acquire data, and methane concentration measurements
with a frequency between $1\,\text{Hz}$ and $10\,\text{Hz}$. The average horizontal spatial resolution is roughly $\frac{v}{f}$ where $v$ is the flight
speed of the drone and $f$ is the measurement frequency. Additionally, drones typically have their own LiDAR (Light
Detection and Ranging) sensor to measure altitude above ground level, a GPS sensor and telemetry relay. The
maximum flight time of a drone is typically limited to about $40\,\text{min}$. More detail on the hardware is given in the
Supplementary Information.

The mass balance method equation, Equation (5), is based on simultaneous information on both the methane
concentration and the wind velocity vector in the flux plane. Hence, it is the best practice to measure the wind
velocity at the position of the drone. Scientific Aviation derive the wind speed at the location of the drone from
the drone's GPS data, the rotor thrust data and the drone's orientation at any moment in time. SeekOps have
recently adopted a similar strategy, but in the historic data discussed in this paper the wind measurements were
taken by a stationary, on-site anemometer at approximately $2\,\text{m}$ above the ground surface; the wind speed at the
drone altitude is derived by applying a wind profile model that has been optimized for the local surface roughness
and drone-derived aerodynamic wind speed.

## 2.4  Numerical simulations

Numerical simulations will be used for a systematic analysis of physical phenomena affecting the error in the mass
balance method. This section describes the methods employed in the numerical simulations.

The coordinate of the drone at each time stamp is simulated based on a curtain or cylindrical flight pattern. The
horizontal spacing between two consecutive measurements is determined by the speed of the drone, frequency of the



measurements and the battery life. In the simulation, for the sake of simplicity this coupling has not been taken into consideration. This means that we simply assume the frequency of 10 Hz .

The methane mass concentration above background in kg/m³ at each location $\mathbf{x}$ is simulated from the Gaussian plume model (Stockie , 2011). In the simulations, either a constant or a time-varying but spatially uniform wind field will be assumed. Using $\xi\eta z$ coordinate frame, the concentration profile downwind of the source $(\xi)$ is then given by:

$$c(\xi,\eta,z) = \frac{\dot{m}}{\pi|\mathbf{u}|\sigma_h\sigma_v}\exp\left(-\frac{\eta^2}{2\sigma_h^2}\right)\left[\exp\left(-\frac{(z-H)^2}{2\sigma_v^2}\right) + \exp\left(-\frac{(z+H)^2}{2\sigma_v^2}\right)\right] \tag{10}$$

where $|\mathbf{u}|$ denotes the absolute value of the wind speed. The first exponent in (10) expresses the Gaussian cross-sectional shape at fixed height. The second exponent illustrates the Gaussian vertical shape at a given $\xi$ which is modified by the third exponent (the ground reflection term). The concentration field is assumed to change immediately with a change of wind. The parameters $\sigma_h$ and $\sigma_v$ denote the standard deviation of the plume in horizontal and vertical direction, respectively. Stockie (2011) explains that these parameters can be tuned to fit a certain situation of atmospheric dispersion. In our present model, we assume that the standard deviations are proportional to the distance from source in the direction of the wind:

$$\sigma_h = \xi\tan(\omega_h), \tag{11a}$$

$$\sigma_v = \xi\tan(\omega_v). \tag{11b}$$

The angles $\omega_h$ and $\omega_v$ denote the opening angles of the plume in horizontal and vertical direction, respectively.

To compute the emission rate using the mass balance Equation (6), a grid is defined with horizontal spacing $\delta P$ and vertical spacing $\delta z$ (equidistant spacing between consecutive points in both directions). The normal to the grid, $\mathbf{n}_{j,k}$ at each point $\mathbf{x}_{j,k}$ is obtained by considering the normal vector to the tangent to the grid at each point $(\mathbf{n}_{j,k}^\perp)$. To calculate the tangent to the grid at each point, the edges of each cell in the grid are considered and the first and last edges are calculated separately based on the type of the grid. The normal vector is then obtained from

$$\langle\mathbf{n}_{j,k},\mathbf{n}_{j,k}^\perp\rangle = 0 \tag{12}$$

The measured methane concentration and the wind field are interpolated onto this grid using nearest neighbor method, referred to as $c_{j,k}$ and $\mathbf{u}_{j,k}$ respectively (Sibson , 1981). The emission rate is obtained by substituting these interpolated values in Equation (6).

We modeled the time-varying wind direction and wind speeds using Ornstein-Uhlenbeck process, Equation (13). This is a random diffusion process that is stationary, Gaussian and Markovian (Kampen , 2007) where the incremental evolution of any random variable $\chi$ is then given by:

$$d\chi = -\frac{(\chi-\mu_\chi)}{\tau}dt + \sqrt{\frac{2\sigma_\chi^2}{\tau}}dW \tag{13}$$



where $\mu_\chi$ is the mean value of $\chi$, and $\sigma_\chi$ is the standard deviation of the probability distribution of $\chi$. The parameter $dW$ is an increment of the Wiener process (e.g., Brownian motion), and $dt$ is the time increment; the current notation is used because the time derivative of the Wiener process, $dW/dt$, is not defined (Kampen, 2007). The physical meaning of Equation (13) is that the Wiener process provides a forcing to the random walk process away from the mean, whereas the first term brings the process back to the mean with a time scale $\tau$. The advantage of using an

Ornstein-Uhlenbeck process for our simulations is that, in the limit of infinitely many Markov steps, the long-term probability distribution of the variable $\chi$ is a stationary Gaussian distribution with mean $\mu_\chi$ and a variance $\sigma_\chi^2$ that is an independent parameter of the problem. The uncertainty in the mass balance calculations can thus be determined as a function of the variability of the wind field.

## 3 Simulation results

In this section, results from the simulation of the 2-D curtain drone flight for different scenarios are presented (similar simulations for cylindrical flight patterns are contained in the appendix). The objective is to assess the importance of various sources of uncertainty on the calculated emission rate. The following sources of uncertainty are considered: a) the choice of data density, i.e., horizontal, and vertical spacing between coordinates, Sections 3.2; b) the angle of the flight lines with respect to the main wind direction, Section 3.3; c) the variation in wind, Section 3.4. In

practice, errors often compound, and there may be a combined effect of multiple errors. However, testing out all permutations and interactions would lead to an intractable number of permutations of parameter settings. It would also complicate the evaluation of which effects have the largest impact on the overall uncertainty, which is the goal of our simulations. Therefore, we limit the analysis to the uncertainty sources presented above.

It is assumed that the concentration field is stationary during the time of the measurements. This might correspond

relate to a constant Gaussian plume travelling horizontally, in a time-averaged horizontal wind field with speeds that can vary as a function of the vertical coordinate. The plume opening angle and the wind field used in the simulation are chosen such that they reflect a situation encountered in practice. A source at location $\mathbf{x}_{j,k}$ with the emission rate of $\dot{m}_{\text{true}} = 5$ kg/h is considered. For the true emission rate $\dot{m}_{\text{true}}$ and the calculated emission rate $\dot{m}$, the relative error in the calculated emission rate is defined as $\epsilon = \frac{\dot{m}}{\dot{m}_{\text{true}}} - 1$.

### 3.1 Ideal case


We will first demonstrate that it is possible to recover the mass emission rate of a source to high accuracy using Equation (6) in an idealised situation. To this end, we consider an ideal case with the following parameters: a) constant wind field with westerly wind with the speed of 5 m/s; b) one flight line passing through the center of the plume; c) the 2-D curtain located at downwind distance of 5 m; d) horizontal and vertical opening angles of

5°; e) equidistant measurements in the horizontal and equidistant measurements in the vertical directions, i.e.,





$\Delta P = \mathrm{const}$ and $\Delta z = \mathrm{const}$; f) flight curtain perpendicular to the wind direction; g) no measurement error in the simulated concentration, the wind field and the location of the drone.

Figure 1 shows the methane concentration as measured during the simulated flat 2-D curtain flight. The vertical and horizontal spacing of $0.3\,\mathrm{m}$ and $0.09\,\mathrm{m}$ are chosen such that a dense grid in both directions is obtained. In practice, it is easier to control flight velocity rather than grid spacing.

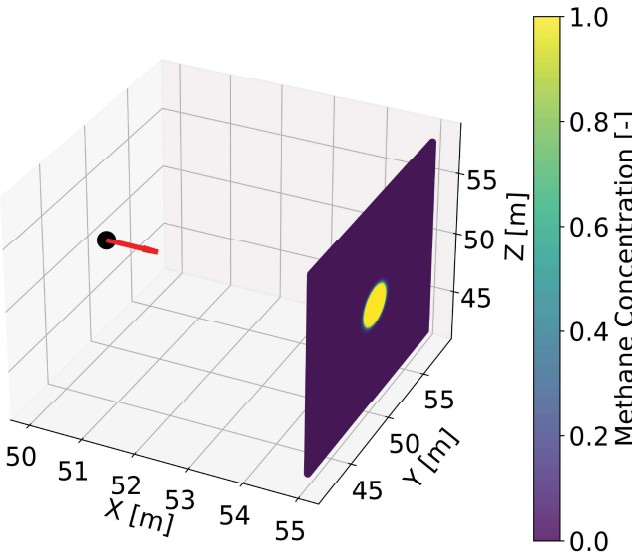

**Figure 1.** 3-dimensional normalized methane concentration above background on a curtain flight located at $5\,\mathrm{m}$ downwind distance of the source (shown by black). The wind velocity is shown with a red vector. The Gaussian plume is assumed to have 5° horizontal and vertical opening angles. There is a horizontal spacing of $0.09\,\mathrm{m}$ between the measurement points and a vertical spacing of $0.3\,\mathrm{m}$ vertical flight lines.

Figure 2 illustrates the concentration measurements in $Pz$ plane along with the calculated emission rate for the source considered in Figure 1 for coarse and fine vertical spacing. The quality of the emission rate estimate degrades with increasing coarseness of the flight lines in the vertical direction, resulting in a deviation of $0.229\,\mathrm{kg/h}$ (4.6% relative error) for a vertical spacing of $1.0\,\mathrm{m}$.

## 3.2 Horizontal and vertical spacings of the curtain drone measurements

### 3.2.1 Non-equidistant vertical spacing

In Section 3.1, the horizontal and vertical spacings between the flight coordinates are assumed to be constant. In practice, however, the assumption of equidistant spacing is easily violated. Here we assess the effect of varying

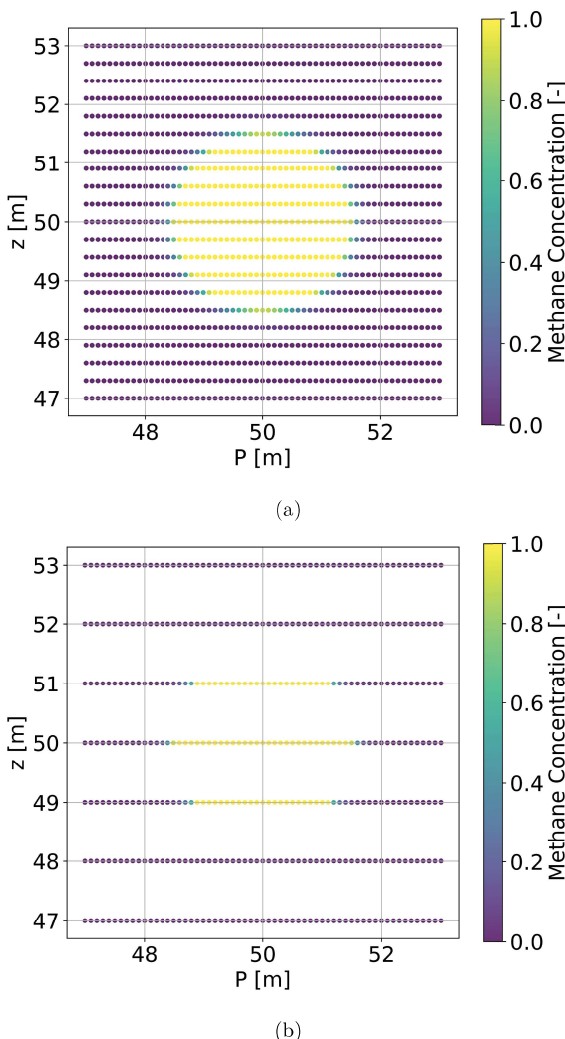

(a)

(b)

**Figure 2.** 2D illustration ($Pz$ plane) of the normalized methane concentration above background simulated using Gaussian plume for a 2-D curtain flight at $5\,\mathrm{m}$ downwind distance from the source, horizontal spacing of $0.09\,\mathrm{m}$ between the coordinates and vertical spacing of a) $0.3\,\mathrm{m}$ with the calculated emission rate equaling $5\,\mathrm{kg/h}$ and b) $1.0\,\mathrm{m}$ with the calculated emission rate equaling $5.229\,\mathrm{kg/h}$.





spacing in the vertical direction on the emission rate error. For $L$ number of flight lines, $L$ random samples from a

Gaussian distribution with zero mean and a standard deviation $\sigma_l$ are drawn as

$$l_i \sim \mathcal{N}(0, \sigma_l^2), \qquad i = 1, 2, ..., L \tag{14}$$

where $\sigma_l^2$ equals a percentage (ranging from 1% to 100%) of the vertical spacing. For a flight line $i$, $l_i$ is added to the simulated vertical coordinates of the drone measurements ($z$). This means the drone measurements on the flight line $i, i = 1, 2, ..., L$ are all shifted equally. For each simulation scenario (i.e., each standard deviation) 300 independent

runs are considered.

Figure 3 shows the box-whisker plot for the emission rate percentage error for non-equidistant vertical spacing. There is no bias in the calculated emission rate with increasing variation in the vertical spacing, but individual measurements cases can have errors up to almost 40% if the standard deviation in the vertical spacing is equal to the nominal vertical spacing. Non-equidistant vertical flight lines introduce the possibility of missing the plume.

Having a larger percentage of the vertical spacing as the standard deviation of the Gaussian distribution, means that the non-homogeneity between the vertical flight lines becomes more pronounced. This means that one might miss the plume by larger extent.

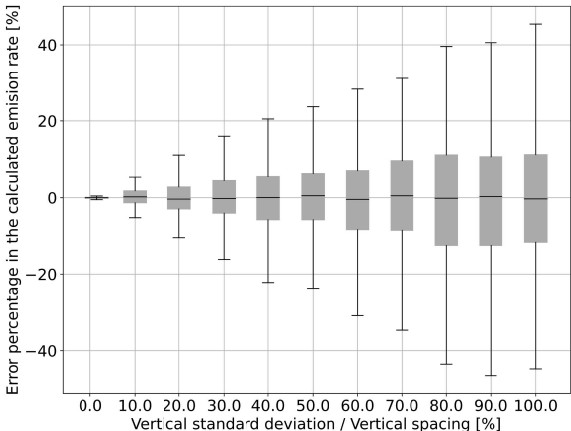

**Figure 3.** Box-whisker plots for the calculated emission rate error for increasing standard deviation of the vertical spacing between flight lines. Here and henceforth, the gray bars denote the 25 %($Q_1$)-75%($Q_3$) quantile range; the lower and upper whiskers show the $Q_1 - 1.5 IQR$ and $Q_3 + 1.5 IQR$ with $IQR$ being the inter-quartile range of the values simulated.

### 3.2.2 Missing the plume center in relation to horizontal and vertical spacings

In Section 3.1, it is assumed that one flight line passes through the centre of the plume in the vertical direction

and there exists a concentration point on this flight line corresponding to the maximum concentration of the plume



on the 2-D curtain. In practice, these assumptions are easily violated as the location of the source is not exactly known. Therefore, here we will assess the potential uncertainty induced by missing the maximum concentration of the plume for different horizontal and vertical spacing of the drone measurements. The resulting error is a function of the horizontal and vertical spacings as the drone is more likely to miss the highest concentration of a plume if the horizontal and vertical spacing is large. To assess the effect of missing the plume, for each horizontal ($\Delta P$) and vertical ($\Delta z$) spacing of the drone measurements, 15 different shift values relative to the plume center (projected on the curtain) ranging from $-\frac{\Delta P}{2}$ to $+\frac{\Delta P}{2}$ in the horizontal and from $-\frac{\Delta z}{2}$ to $+\frac{\Delta z}{2}$ in the vertical directions are considered. For each shifted location, the emission rate error is calculated, i.e., 225 ($15 \times 15$) instances of calculated emission rate errors. For all the locations except the one where one flight line passes the centre of the plume, the maximum concentration is missed. Therefore, the emission rate is usually underestimated and the maximum error representing the worst situation is considered for each combination of horizontal and vertical spacings.

For a plume with a known opening angle in the horizontal and vertical directions ($\omega_h$ and $\omega_v$), the calculated emission rate remains unchanged for varying downwind distance $d$ if the horizontal and vertical spacings are scaled according to the opening angle of the plume. In order to have a generic representation of the maximum error for different horizontal and vertical spacings and a downwind distance $d$, the horizontal ($\Delta P$) and vertical ($\Delta z$) spacings can be expressed as factors of the plume horizontal and vertical standard deviations, $\sigma_h$ and $\sigma_v$ in Equation (11b), respectively as the following dimensionless quantities:

$$\Delta P' = \frac{\Delta P}{\sigma_h} = \frac{\Delta P}{d \tan \omega_h}, \tag{15a}$$

$$\Delta z' = \frac{\Delta z}{\sigma_v} = \frac{\Delta z}{d \tan \omega_z}, \tag{15b}$$

Shown in Figure 4 is the maximum error in the calculated emission rate for different dimensionless horizontal and vertical spacings for a curtain flight located 5 m downwind distance of a source with the emission rate $\dot{m} = 5$ kg/h. As seen, the estimated emission rate error decreases with higher frequency (i.e., smaller horizontal spacing) and smaller vertical spacing. It should be highlighted that even with a high frequency drone flying with a small vertical spacing, the error will not reach zero due to the existence of other error contributors.

### 3.3 Angle between flight lines and wind direction

In the ideal situations described in Section 3.1, the drone flies in a direction perpendicular to the wind field. However, in real-world scenarios such a flight design might be impractical due to the existence of buildings and obstacles, or safety requirements. It is thus relevant to assess how the deviation of the flight curtain from the direction perpendicular to the wind field affects the calculated the emission rate. Figure 5 shows the result as a function of the angle of the curtain with respect to the wind direction, for three different values of the plume opening angle. As seen, the error is negligible for situations in which the deviation of the 2-D curtain from the direction perpendicular to the wind field is less than 45°. If the flight pattern is not perfectly perpendicular to the wind direction, an



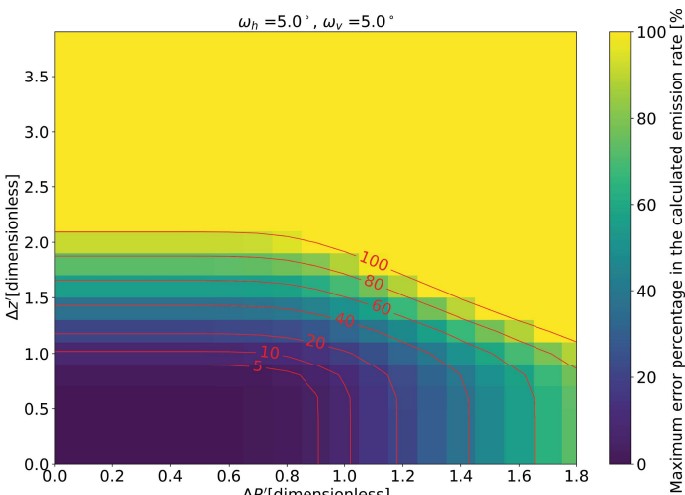

**Figure 4.** Maximum emission rate error in percentage (over 225 experiments corresponding to missing the plume by different percentages of the horizontal and vertical spacing) for varying dimensionless horizontal and vertical spacings. Shown with red are the error contours.

error term emerges due to the effect of diffusion, see first term in the right-hand side of Equation (4). In the mass balance equation, Equation (5), the diffusion term is discarded and only the advection term is considered. For the non-perpendicular flight patterns, however, the term $\mathbf{n} \cdot \mathbf{e}_y \frac{dc}{dy}$ in Equation (4) is not exactly equal to zero (here, $\mathbf{e}_y$ denotes the unit vector pointing in the $y$-direction). For a westerly wind field, the deviation in the northern part of the plume may, to first order, be compensated by opposite deviations in the southern part of the plume, but the deviations can become considerable at large measurement angles. The actual contribution of the error from this source is proportional to the value of the diffusion coefficient $K$, and it leads to a slight over-estimate of the mass emission rate that increases with the opening angle of the plume (usually the opening angle is proportional to $K$). If the flight trajectory deviates from the direction perpendicular to the wind by more than $\approx 70°$, underestimation occurs and the magnitude of underestimation increases with the deviation angle. The main cause for this error is that part of the diverging plume will never be captured by the vertical curtain; if the opening angle is larger, a larger percentage of the plume will be missed in this way and the error thus increases.

### 3.4 Time variations in wind speed and/or wind direction

Another assumption considered in Section 3.1, was that that the wind field is time-invariant, however, in practice this assumption is rarely true. To take this issue into account, we consider a 5 m/s westerly wind at the start of the flight and randomly draw samples from an Ornstein-Uhlenbeck process with a defined standard deviation in



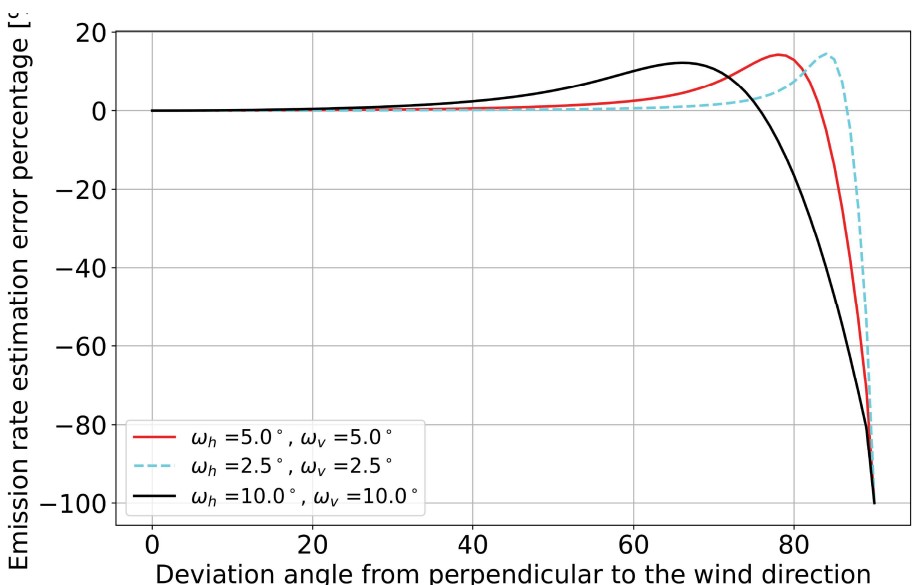

**Figure 5.** The emission rate error percentage as a function of deviation of the 2-D curtain from the direction perpendicular to the wind velocity obtained from the simulation. Results are shown for three different opening angles of the Gaussian plume: 2.5°, 5.0° and 10.0°.

the wind speed and/or the wind direction to simulate a time-varying wind field. For each standard deviation, 300 independent runs are considered, and the results are shown for time-varying wind direction (a), wind speed (b) and their combination (c) in Figure 6. Figure 6a) shows that there is no bias in the error in the calculated emission rate, but individual calculations can be erroneous, especially if the variation in the wind direction is large. As for time-varying wind speed, increases in instantaneous variations up to 30% of the wind speed do not seem to have a noticeable effect (Figure 6b); larger variations lead to underestimation of the emission rate. However, compared to the impact of time-varying wind direction, variations in wind speed play a relatively small role. The error due to time varying wind field (see Figure 6c) is dominated by the variations in the wind direction.

### 3.5 Other potential sources of error

There are other potential sources of error and uncertainty, such as the precision of the methane concentration sensor, uncertainty in GPS measurements, and the uncertainty of wind velocity in the turbulent atmospheric boundary layer between the source and the vertical measurement curtain during the entire time required to complete the data acquisition. The detailed properties of the emission source, such as the temperature of the emitted gas and its buoyancy, can also play a role in how a methane plume moves through the atmosphere. Finally, time variation in




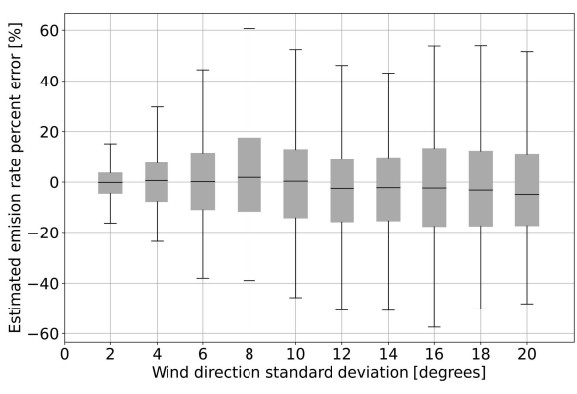

(a)

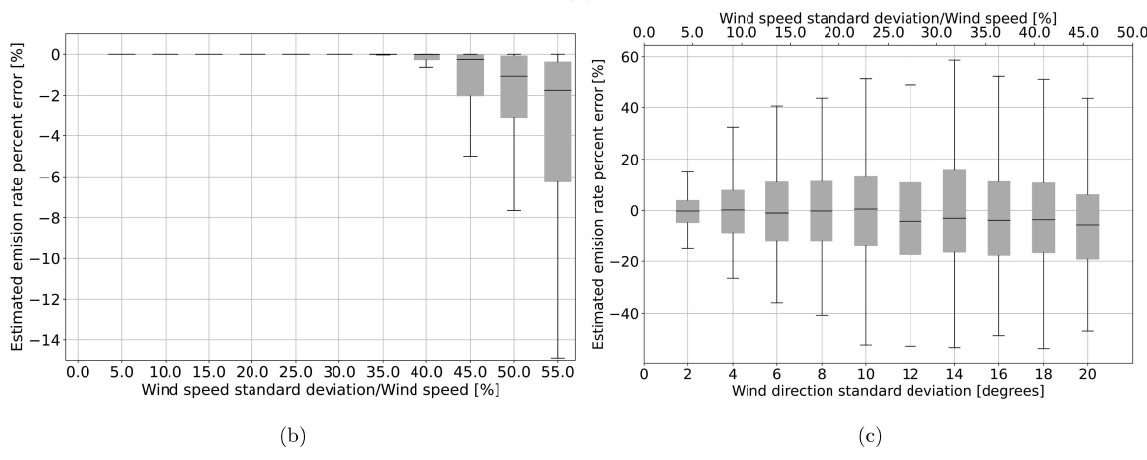

(b)                                                              (c)

**Figure 6.** Box-whisker plots for the calculated emission rate error for increasing standard deviation of a) time-varying wind direction, b) time-varying speed and c) time varying wind velocity (both direction and speed at the same time). The 2-D curtain is located 5 m/s downwind from the source.



mass emission rates from sources are not considered in this paper, but they may well be a source of uncertainty in practice in site-level measurements under the OGMP framework.

## 330  4  Real-life examples

### 4.1  Historic controlled release data from Scientific Aviation

Nine controlled release experiments from Scientific Aviation were analyzed to gain a better understanding of the empirical uncertainty of the mass balance method, and a more detailed appreciation of the underlying assumptions that go into the computation of the methane emission rate. The experiments were carried out near the Scientific Aviation office in Boulder, Colorado, in 2019 and 2020.

335

Figure 7a shows the flight path of measurement "A", done on 13 November 2019. The dots, which indicate the flight trajectory, are colored by the methane concentration measured. Figure 7b shows the altitude of the drone in the course of time during measurement "A". Apparently, multiple transects of the methane plume were attempted at different altitudes. Figure 7c shows a time series of the methane concentration measurements.

Although the objective was to fly a vertical curtain pattern, comparing the left graph of Figure 7 to Figure 1 highlights the difference between the simulated drone measurements and actual drone measurement by a pilot. Some of the assumptions considered in Section Section 3.1 are likely violated: there is no guarantee that there will be one flight line passing the center of the plume, or that the horizontal and vertical spacings between adjacent concentration are nearly equidistant. The time series of concentration measurements and altitude indicates that there are upward

345

(solid red) or downward (dashed black) curtains where the plume is missed. It is thus important to assess how the emission rate calculated from the mass balance method is affected by irregularities in the measurements taken by the drone.

As an illustration, we will consider the data from flight "A"in more detail. If we take into account all the concentration measurements, except those corresponding to the drone flying to the location of interest, we can use

350

a nearest neighbor algorithm with the euclidean distance metric to project the concentration measurements onto a vertical plane, other distance metrics such as Chebyshev and Minkowski distances can be used. We can then compute the total emission rate with the mass balance method using Equation (8), where the total mass emission rate is the average of multiple individual curtain patterns. The calculated emission rate using all the concentration measurements, without any post processing, is $55.7\,\mathrm{g/h}$. This is 80% higher than the true release rate of $30.8\,\mathrm{g/h}$.

355

If, on the other hand, we choose a different postprocessing method to calculate the emission rate, the results can be very different. For example, we can subdivide the flight trajectory in measurement "A"into 2 upward curtains, indicated as 1 and 4 with red rectangles in Figure 7b, and 2 downward curtains, indicated as 2 and 3 with black rectangles. Shown in Figure 8 are the 3-dimensional concentration measurements for each of these curtains. The curtains have been chosen such that they contain elevated concentrations and they do not contain conflicting concentration

360

values at the same altitude. Presented in Table 1 is the calculated emission rate for each individual upward and



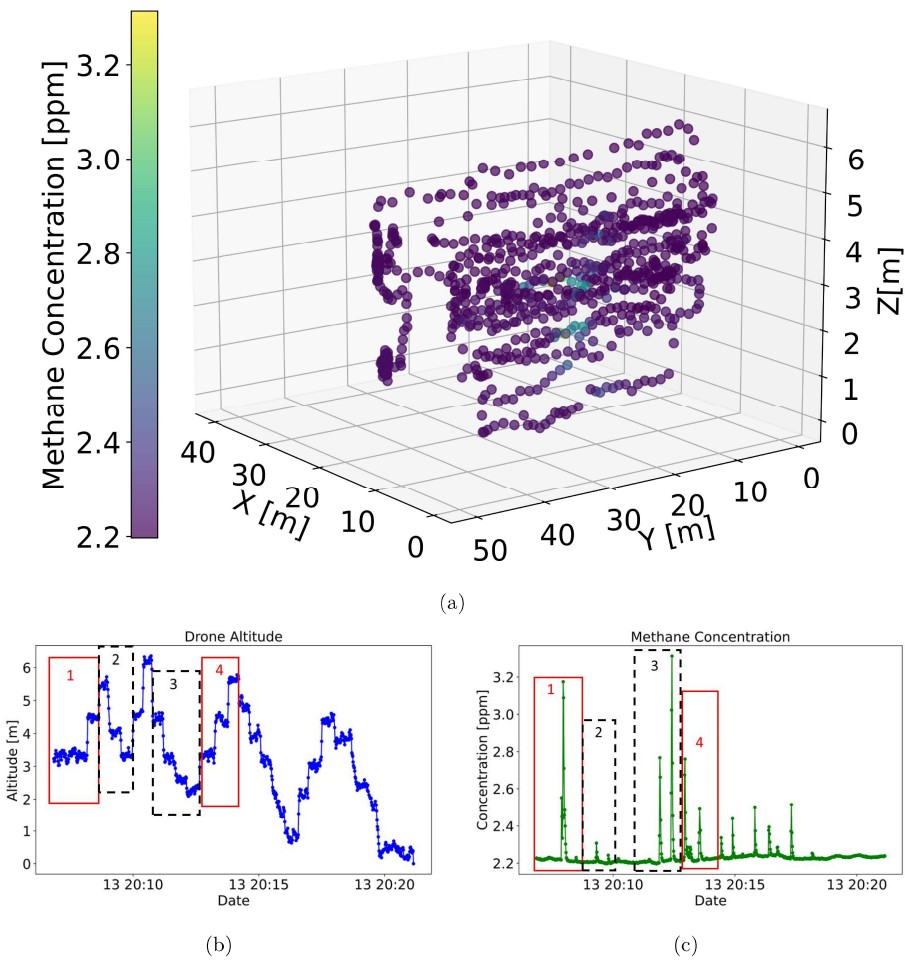

**Figure 7.** a) Methane concentration data plotted against the location of measurement in a 3-dimensional space. b) Time series of the drone altitude, with the first upward transect denoted by "1"(solid red), the first downward transect denoted by "2"(dashed black); the second downward transect denoted by "3"(solid red), the second upward transect denoted by ""4""(dashed black),c) Concentration time series for the controlled release data "A"acquired on 2019-11-11.



downward curtain using Equation (8) along with the overall average emission rate from the curtains selected. The result is an emission estimate of 28.26 g/h, which is only 8.2% lower than the true emission rate. This shows that the postprocessing method can potentially decrease the error in the calculated emission rate considerably. Adopting more sophisticated interpolation methods, such as a Gaussian kernel smoother or Gaussian process, Rasmussen
365 (2004) and Monaghan (2005), might further improve the agreement between the calculated emission rate and the true release rate.

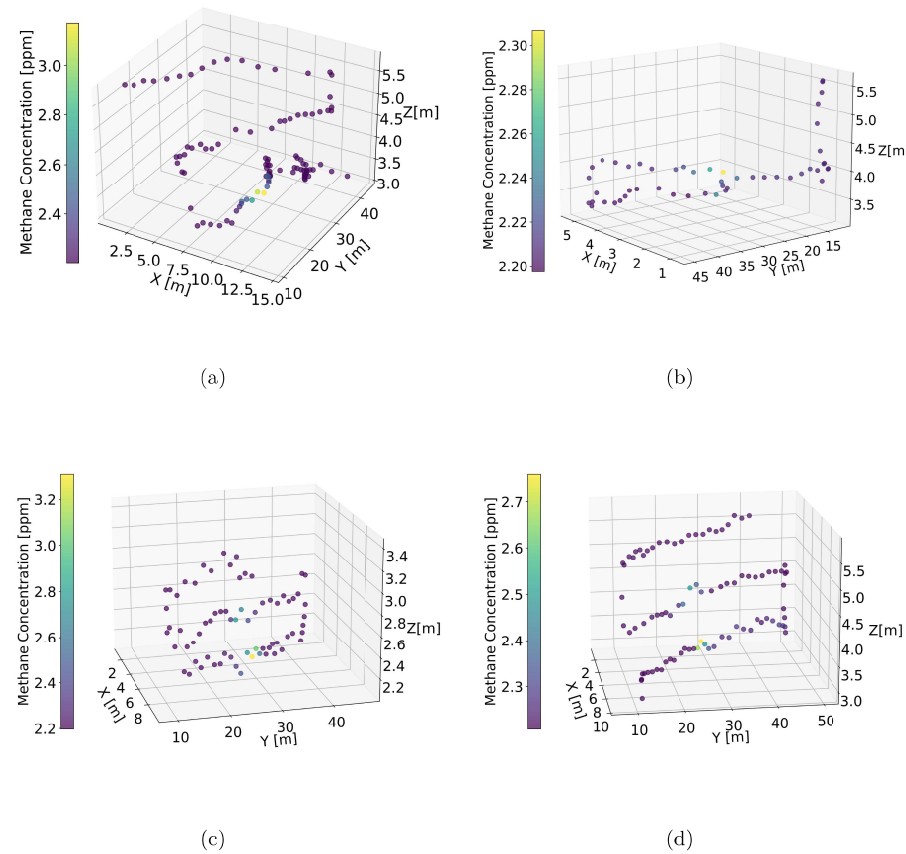

**Figure 8.** 3-dimensional concentration measurements for a) upward curtain "1", b) downward curtain "2", c) downward curtain "3"and d) upward curtain "4".



**Table 1.** Emission rates calculated per curtain and the average emission rate between upward and downward curtains for the measurement "A", which was carried out on 13 November 2019. The emission rates in the table are calculated using Equation (8). The true emission rate was $30.8\,\mathrm{g/h}$.

| Curtain | Emission rate [g/h] | Average between U and D | Emission rate error [%] |
|---|---|---|---|
| Upward (U) 1 | 14.48 | | |
| Downward (D) 2 | 12.30 | 13.39 | -56 |
| Downward (D) 3 | 46.73 | | |
| Upward (U) 4 | 40.10 | 43.14 | 43 |
| Average from the four selected curtains | 28.26 | | -8.2 |
| Average from all curtains | 55.7 | | +80 |

If we employ Equation (9) for the computation of the methane emission rate, then the result is $34\,\mathrm{g/h}$, which is about 10% higher than the true answer of $30.8\,\mathrm{g/h}$. The difference between the two calculation methods is due to a different interpolation methodology and post processing technique. Since the curtains are not equal in this case (some curtains span a larger range of altitudes than others), the latter method (using Equation (9)) gives a different result than the average of the mass emission rates computed per curtain (using Equation (8)).

Shown in Table 2 is the calculated emission rate from Equation (8) and Equation (9) along with the true release rate for nine different controlled release experiments. The last column of Table 2 shows the dimensionless vertical spacing $\Delta z'$, computed from the flight data for each controlled release survey using Equation (15b). For want of detailed long-term wind data in this case, we assumed the plume opening angle $\omega_z$ to be equal to $5°$ in all cases. The values in the last column show that the dimensionless vertical spacing $\Delta z'$ was smaller than 1 in most cases, and never exceeded 1.4; looking back at the contour plot in Figure 4, it is clear that the potential avoidable error due to the choice of vertical spacing was relatively low in all cases. Although, as noted earlier, this does not mean that there could not have been other sources of error, at least these are promising measurements with a small overall error if an appropriate postprocessing method is selected to process the measured concentration and wind data.

The results in Table 2 illustrate that indeed there is, in general, reasonable agreement between the calculated mass emission rates and the true methane emission rates. The root-mean-square of the error percentage is 31.0% when using Equation (8), and it is 28.9% when using Equation (9). On individual cases, however, there can be a considerable difference between the two methods, see for example case C (62.54% error versus -13%) or case H (6.63% error versus 29%). There does not appear to be a systematic bias that favors one method or the other in all the cases.

A tentative conclusion from the controlled release data is that the "best"data analytic method depends primarily on how the data was collected. For instance, Equation (8) appears to work best in cases where the data is collected in one full curtain with sufficiently small and constant vertical and horizontal spacings, with a fairly constant wind direction during the flight. In a second or third flight, the same curtain pattern may be flown, with again enough data



**Table 2.** Calculated emission rate and the error percentage for the controlled release experiments carried out by Scientific Aviation. Mass emission rates are computed from the data using Equation (8) and Equation (9), described in Section 2.3. The value of the dimensionless vertical spacing $\Delta z'$ is shown in the right-most column.

| Name | Flight type | $\dot{m}_{\text{true}}$ g/h | $\dot{m}$ from Eq. (8) g/h [$\epsilon\%$] | $\dot{m}$ from Eq. (9) g/h [$\epsilon\%$] | $\Delta z'$ |
|------|-------------|------|------|------|------|
| A | Curtain | 30.8 | 28.26 [-8.24%] | 34 [10%] | 0.57 |
| B | Curtain | 31.2 | 28.65 [-8.17%] | 39 [25%] | 0.91 |
| C | Curtain | 20.8 | 33.81 [62.54%] | 18 [-13%] | 0.43 |
| D | Cylinder | 20.04 | 20.06 [0.1%] | 19 [-5%] | 1.2 |
| E | Curtain | 10.02 | 3.56 [-64.47%] | 3 [-70%] | 0.7 |
| F | Cylinder | 10.15 | 9.64 [-5.02%] | 7.9 [-22%] | 1.07 |
| G | Cylinder | 30.45 | 34.78 [14.22%] | 32.5 [7%] | 0.97 |
| H | Cylinder | 14.46 | 15.42 [6.63%] | 18.6 [29%] | 1.4 |
| I | Cylinder | 12.35 | 10.7 [-13.36%] | 10 [19%] | 1.2 |
| | RMS percentage error | | 31.0% | 28.9% | |

for a reliable mass balance calculation per flight. In that case, Equation (8) is probably more appropriate because it uses the full mass balance method based on a direct discretization of the boundary integral (Equation (5)). We can see from Table 2 that this approach seems to work well for cylinder-shaped flight patterns in particular. On the other hand, if the data acquisition shows multiple curtains with a small number of data points per curtain (as the flight pattern from measurement "A" in Figure 8, for instance), then it is wise to group the measurements together in bins per altitude in order to collect sufficient statistics to base the mass balance calculation on, using Equation (9). If Equation (8) is used, it may be necessary to manually select useful data from less useful data – as was illustrated earlier in Table 1 for flight "A".

## 4.2 Historic measurements from SeekOps

To understand how the uncertainty in methane emission rate quantification translates into potential errors in practice, a total number of 1,001 historical drone flights from SeekOps were analyzed (all the data was anonymized with regards to the geographical location and the paying customer). The data from the flights contained the position of the drone and the concentration measured at that moment in time; in addition, simultaneous data from the ground-based anemometer (at an altitude of approximately 2 m) was provided, which was extrapolated to the altitude of the drone using a predefined velocity profile from turbulent flow theory. This extrapolation itself can be significant source of uncertainty in practice, because the actual wind velocity at the position of the drone can be very different from the wind velocity measured close to the ground. Despite this caveat, we use the data from anemometer to obtain a first estimate of the variability of the wind conditions. In summary, the following parameters were calculated from the data collected during each flight: a) from the anemometer data: the wind average speed, wind standard deviation



in North-South and East-West directions; b) from the flight pattern: the average horizontal and vertical spacing in methane concentration measurements, and standard deviations of these quantities; c) from site layout: the downward distance from source, estimated from the source location in an equipment area. In order to allow the uncertainty analysis as explained in Section 3, the raw wind data needs to be converted into an absolute wind speed and its root-mean-square fluctuation, and a plume opening angle. The conversion process is described in the Supplementary

Information.

With these parameters calculated, the potential errors were calculated for each of the 1,001 historic flights by SeekOps. Based on the available information, the errors in the emission rate calculation due to the following sources are calculated for each flight: a) not passing the center of the plume due to large vertical spacing between the flight lines; b) non-equidistant vertical spacing; c) time-varying wind speed; d) time-varying wind direction.

We will report the potential errors due to these parameters for flights in four categories, based on the plume opening angle at the time of measurement. The vertical plume opening angle in all 1,001 flights analysed covers the full range of angles from 0° to 90°; for visualization purposes, we will report the potential errors in 4 groups of the plume opening angles: between 0° and 22.5° (with an average opening angle of 10.7° for measurements with a plume opening angle in this category); between 21.4° and 45° (with an average of 30.5°); between 45° and 67.5° (with an

average of 50.8°); and between 67.5° and 90° (with an average of 71.1°).

**Not passing the center of the plume**

The results for the effect of non-zero horizontal and vertical spacings in the methane concentration measurements are shown in Figure 9. Figure 9a shows the results for the category of flights where the plume opening angle was relatively narrow. Color shades from blue to red indicate increase in the errors in the calculated emission rate. The

dimensionless horizontal ($\Delta P'$) and vertical ($\Delta z'$) spacings are determined from the flight data provided by SeekOps; these coordinates are then plotted as red crosses onto the contour plot that was determined for a plume with opening angle of 10.7°. Most red crosses are located close to the origin of the graph. This area corresponds to low potential errors due to the choices of dimensionless spacings. There are also flights, however, where the dimensionless spacing was larger than 1 (in horizontal and/or in vertical direction), and for those cases the errors become potentially very

large: more than 100% of the measured value. The reason why there is a significant number of data points in this region may actually be due to unexpectedly coherent wind situations: if the wind is steady and turbulent fluctuations are low, then the plume opening angle is small, and the dimensionless horizontal and vertical spacings can become large since the opening angle is in the denominator of Equation (15b) (through Equation (11b)). There is thus the risk that the plume passes through the flight lines, meaning that the drone misses the plume altogether.

The results for the wide plume opening angles are shown in the other graphs of Figure 9. In these cases, we see that most data points from SeekOps flights are located near the origin in regions where the expected error is relatively low.

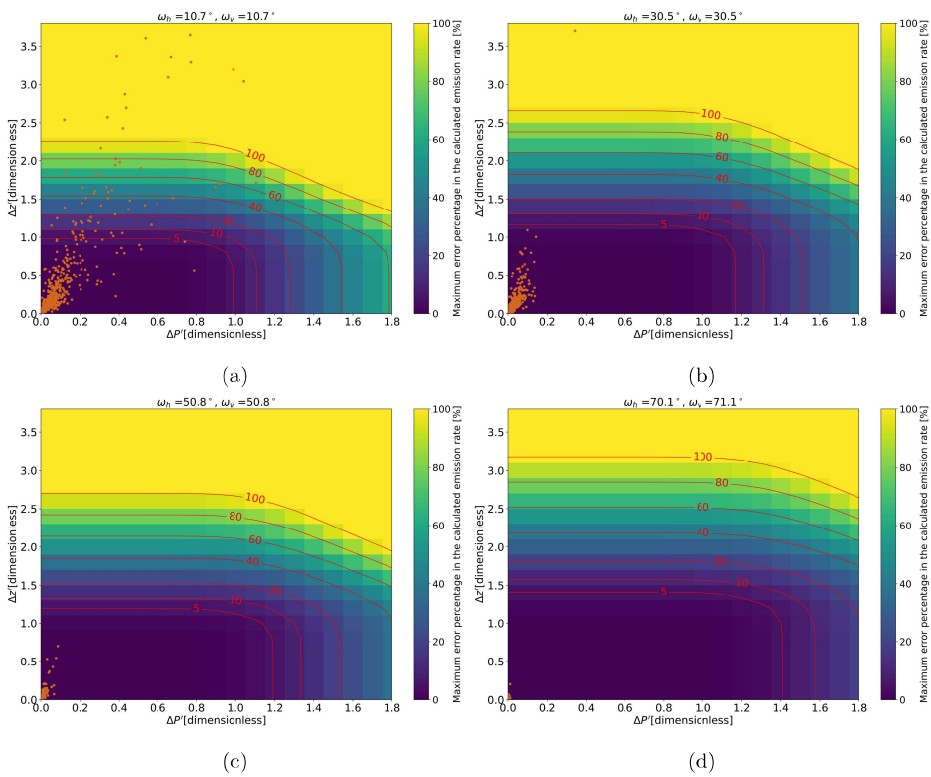

**Figure 9.** Maximum calculated emission rate error (over 225 calculations corresponding to missing the plume by different percentages of the horizontal and vertical spacing) for varying dimensionless horizontal and vertical spacings for four different groups of plume opening angles: a) plume opening angle of 10.7°, b) plume opening angle of 30.5°, c) plume opening angle of 50.8°, and d) plume opening angle of 71.1°. The orange dots indicate the horizontal and vertical spacings for four different groups of plume opening angles a) between 0° and 22.5°, b) between 21.4° and 45°, c) between 45° and 67.5° and d) between 67.5° and 90°. Shown with red are the error contours.





**Non-equidistant vertical spacing**

Now we analyse the errors that can potentially occur due to the standard deviation of vertical spacings. The result
is shown in Figure 10, for each of the four categories of plume opening angles. For each flight, the location on the
horizonal axis is calculated using the vertical spacing and its standard deviations and shown with a red cross. The
box-whisker plot is generated from the numerical simulations described in Section 3, using the average horizontal and
vertical spacings, vertical opening angles and downwind distance over 300 runs. For each run, the flight coordinates
are generated by varying the flight lines randomly using a Gaussian distribution. The green bars are a histogram of
the incidence rate in the 1,001 flights of the ratio between the vertical standard deviation and the vertical spacing.
Figure 10a shows the results for the category of flights where the plume opening angle was relatively narrow. The
results for the wide plume opening angles are shown in the other graphs of Figure 10. For most flights, the variation
in the vertical spacing is less than 10% of the vertical spacing. Therefore, this effect is not expected to be a major
source of error or uncertainty in the measurement.

**Time variation in wind speed and/or wind direction**

We will now look at the errors that can potentially occur due to time-varying wind speed and wind direction, shown
in Figure 11 and Figure 12 respectively. For each flight, the location on the horizonal axis is calculated using the
wind speed, and wind direction, and their corresponding standard deviation. The errors bars are calculated from the
numerical simulations described in Section 3, from the average over 300 independent runs per flight. For each run,
the wind speed and wind directions are generated from the Ornstein-Uhlenbeck process. The box-whisker plot is
generated using the average wind speed, wind direction, their corresponding variations and vertical opening angles.
   Figure 11a shows the results for time-varying wind speed and for the category of flights where the plume opening
angle was relatively narrow. As the plume opening angle gets wider and the variation of the wind speed increases,
the error percentage in the calculated emission rate increases. It can also be seen across all graphs in Figure 11 that
the higher opening angles typically correspond to higher variation in the wind drection, see e.g. the horitontal axes
in Figure 11c) and Figure 11d).
   Figure 12a shows the results for time-varying wind directions for the category of flights where the plume opening
angle was relatively narrow. The results for the wide plume opening angles are shown in the other graphs of Figure 12.
Most flights have wind direction variability less than 40° and for increasing wind direction variability, the absolute
error increases linearly.
   By means of summary, Figure 13 shows the cumulative statistics over all 1,001 flights with regards to the theoretical
error percentage in the emission rate, for each of the different sources of errors discussed above. Not passing the
center of the plume has the largest contribution, as was already observed in Figure 9a). About 55% of the flights
could have more than 10% emission rate percentage theoretical error because of missing the center of the plume.
The other sources of error usually have a smaller impact. For instance, all the measurements have less than 10%



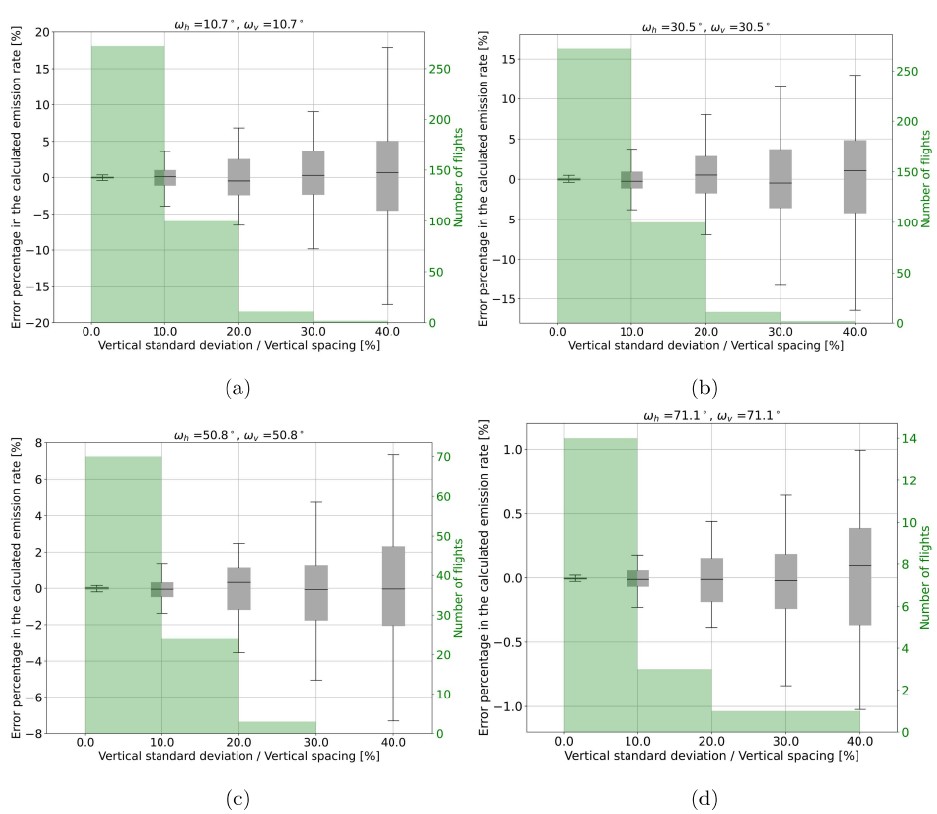

(a)  (b)

(c)  (d)

**Figure 10.** Theoretical box-whisker plots due to non-equidistant vertical spacing for four different groups of plume opening angles. a) plume opening angle of 10.7°, b) plume opening angle of 30.5°, c) plume opening angle of 50.8°, and d) plume opening angle of 71.1°. The green bars show the histogram of the actual flight data as a function of the vertical standard deviation divided by the vertical spacing for four different groups of plume opening angles a) between 0° and 22.5°, b) between 21.4° and 45°, c) between 45° and 67.5° and d) between 67.5° and 90°.

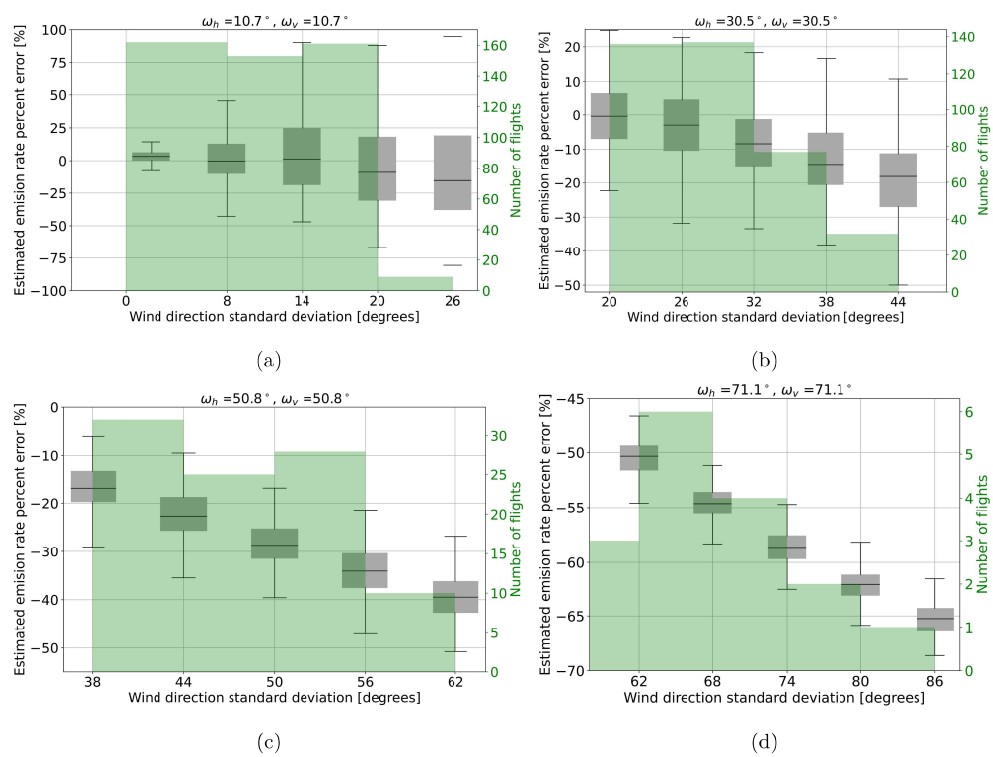

**Figure 11.** Potential box-whisker plots to time-varying wind direction. a) plume opening angle of 10.7°, b) plume opening angle of 30.5°, c) plume opening angle of 50.8°, and d) plume opening angle of 71.1°. The green bars show the histogram of the actual flight data as a function of the vertical standard deviation divided by the vertical spacing for four different groups of plume opening angles a) between 0° and 22.5°, b) between 21.4° and 45°, c) between 45° and 67.5° and d) between 67.5° and 90°.



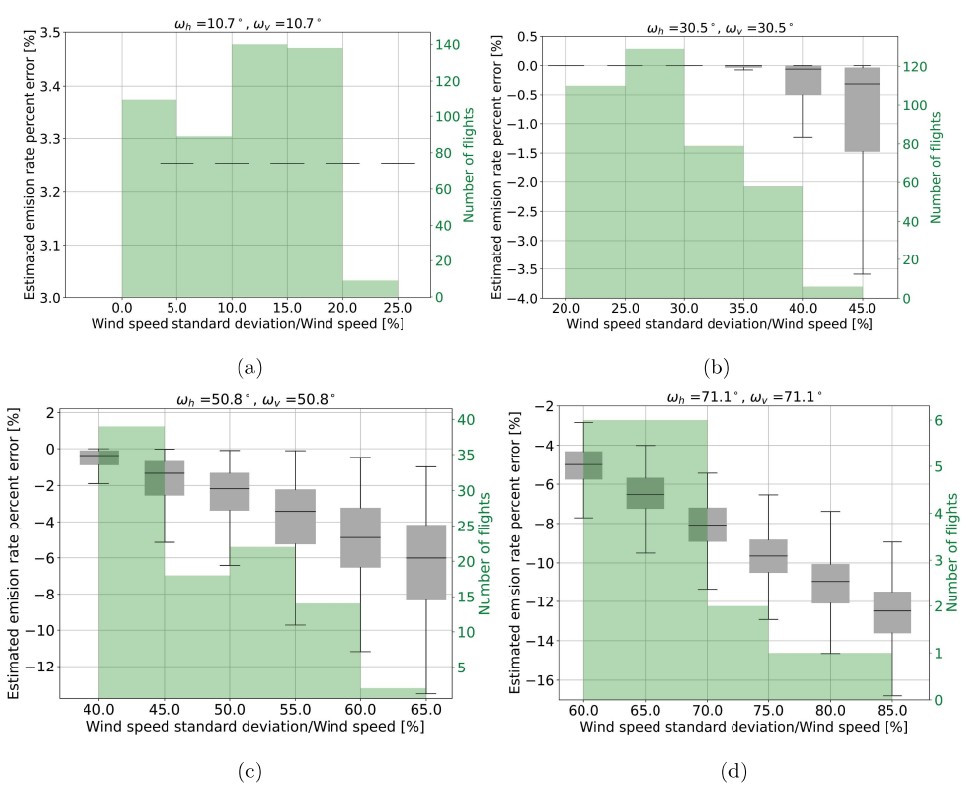

**Figure 12.** Potential box-whisker plots due to time-varying wind speed. a) plume opening angle of 10.7°, b) plume opening angle of 30.5°, c) plume opening angle of 50.8°, and d) plume opening angle of 71.1°. The green bars show the histogram of the actual flight data as a function of the vertical standard deviation divided by the vertical spacing for four different groups of plume opening angles a) between 0° and 22.5°, b) between 21.4° and 45°, c) between 45° and 67.5° and d) between 67.5° and 90°.





theoretical error due to non-equidistant vertical flight lines. For the contribution of the time-varying wind speed, around 55% of the flights have less than 10% error in the calculated emission rate; the effect of time-varying wind direction is expected to be larger than 10% only in 20% of the cases. Around 70% of the flights have the emission rate error less than 50% due to not passing the center of the plume. Between 97% and 100% of the flights have

less than 50% theoretical error in the calculated emission rate due to time-varying wind direction and speed and non-equidistant vertical flight lines.

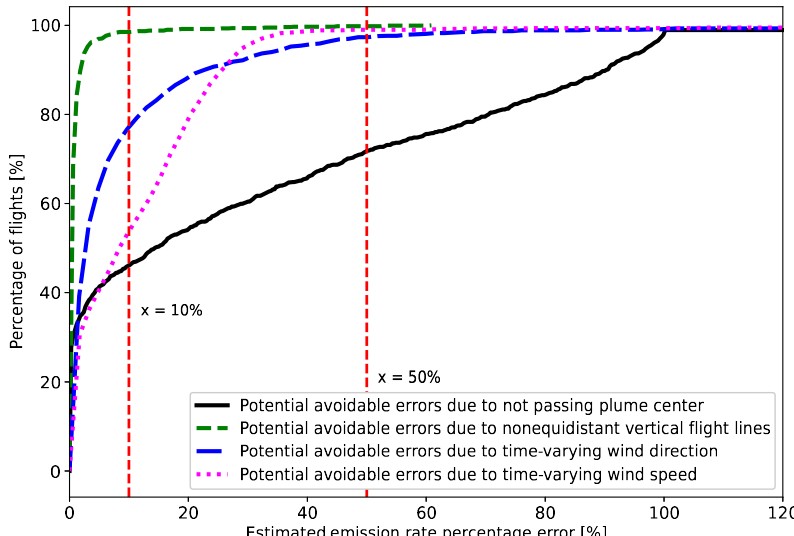

**Figure 13.** Cumulative distribution of flights (totalling 1001) as a function of theoretically calculated maximum error for potential avoidable error percentage due to not passing the plume centre (solid black), non-equidistant vertical flight lines (dashed green), time-varying wind directions (dotted dashed blue) and time-varying wind speed (dotted pink). Vertical dashed red lines indicate 10% and 50% error in the estimated emission rate.

## 5 Discussion

The theoretical framework presented above allows for a post-hoc analysis of historic data, which allows the evaluation of the quality of the measurement from a methane emission quantification survey. Potential errors and uncertainties

in the measurements can be identified. With this, it is possible to back-calculate errors and uncertainty in the results from historic measurement data.



The analysis on historic flight data from Scientific Aviation in controlled release experiments, reported in Section 4.1, shows that there can be a wide range of outcomes in the mass balance method for different curtain flights – even during the same controlled release event. For the best result, the data analysis procedure must be adapted
to the measurements taken.

The theoretical analysis can also be used to evaluate the quality of a historical data set of a large number of surveys. From the analysis reported in Section 4.2, it seems that the largest risk of uncertainty is caused by too large of a vertical spacing of the flight lines in combination with a relatively short downwind distance from the source. The uncertainty is highest in cases when the wind field is steady and the plume opening angle is small –
contradicting some 'conventional wisdom' in the industry that the uncertainty is always lowest when the wind field is steady. What we find is that this assumption is only true if the measurements are taken at a sufficient distance downwind of a source with a sufficiently small vertical spacing between flight lines.

In addition to this risk of missing the methane plume in the case of steady winds, it becomes also clear from our analysis (Figure 12c and Figure 12d, in particular) that large fluctuations in the wind direction can result in a
significant error. Such a situation typically occurs when the atmosphere is unstable or when the average wind speed is very low. In these cases, the fast diffusion of the methane plume can easily result in parts of the plume being missed; indeed, Figure 12c and Figure 12d show that the methane emission is usually underestimated by the mass balance method in these cases.

Our observations can also provide guidelines to future campaigns. The two main recommendations based on
the present work are: a) For each individual quantification survey, fly multiple curtain patterns and choose the appropriate data analysis method to avoid excessive influence of unrepresentative data (outliers) on the result. b) Take measurements with a sufficiently small vertical spacing and at a sufficiently large distance downwind of the source for the wind conditions on the day. In particular, the dimensionless horizontal and vertical spacings $\Delta P'$ and $\Delta z'$ have to be smaller than 1 for the best result (see Equation (15b) for the full expression). There is an optimum
though: downwind distance should not be so large that methane plumes from the facility of interest are missed, and measurements very far downwind are also unfavorable because of the lower methane concentrations expected, which will lead to a lower signal-to-noise ratio. It appears to us that values of $\Delta P'$ and $\Delta z'$ between 0.1 and 1.0 are optimal. To make this guideline more practical: if we assume that most opening angles are larger than 5 degrees, which is the case in about 98% of the 1,001 onshore drone flights considered in this paper, then the guidance is
that the vertical and horizontal spacings have to be smaller than 1/10-th of the downwind distance to the source; in offshore measurements, this guidance may have to be adapted to address different atmospheric stability over open water.

As a final reflection, we would like to emphasize that we have not focused in this paper on measurement uncertainty of the sensors. This means that we do not account for possible deviations due to finite precision in the methane
concentration measurements, nor do we assume any error in the wind measurement. In addition, our analyses are based on the assumption that the wind vector at the position of the drone is determined perfectly. In reality, of



course, this is never the case, and there are various methods deployed by different service providers to measure the wind. We only observe here that any error in the wind measurement is likely to propagate in the computations in the mass balance method as per Equation (6). It is not wise to assume that any error in wind measurement will

somehow be compensated for by other errors. Notwithstanding other sources of error, we believe that this paper provides some guidelines to minimize avoidable errors that can result from choices in the data acquisition process and in the methodology used for postprocessing.

## 6 Conclusions

Site level measurements of methane emissions are often used by operators for reconciliation with bottom-up emission

inventories with the aim to improve accuracy, thoroughness and confidence in reported methane emissions. In that context it is of critical importance to minimize measurement errors and to understand the associated uncertainty. This paper describes a systematic analysis of potential errors in methane emission quantification surveys using the mass balance method, for parameters related to the acquisition of concentration data and the method of postprocessing. The analysis is applied to a quadcopter drone with a high-precision methane sensor flying in a vertical curtain

pattern; the total mass emission rate can then be computed post hoc from the measured methane concentration data and simultaneously measured wind data. We find that the most important source of potential error can be expressed as a dimensionless number with the spatial resolution of the methane concentration measurements and the downwind distance from the main emission sources. The potential error is largest (and, indeed, can be as high as 100%) in situations where the wind speed is steady and the methane plume has a coherent shape – contradicting

the intuition of some operators in the industry.

The learnings from our theoretical error analysis have been applied to a number of historical measurements in a controlled release setting. We show how the learnings on the main sources of error can be used to eliminate potential errors during the postprocessing of flight data; we show that the reported results can be very close to the actual methane emission rates if the appropriate data analysis method is selected. Second, we have evaluated an aggregated

data set of 1,001 historical drone flights. Our analysis shows that the potential errors in the mass balance method can be of the order of 100% on occasions, even though the theoretical error from the identified error sources were relatively small in the majority of the flights considered.

The discussion section provides some guidelines on how to avoid or minimize potential errors in drone measurements for methane emission quantification. The two main recommendations are to (1) fly multiple curtain patterns

in each individual survey in combination with an appropriate data analysis method to avoid excessive influence of unrepresentative data (outliers) on the result, and (2) to take measurements with a sufficiently small vertical spacing and at a sufficiently large distance downstream of the source for the wind conditions on the day, with the dimensionless horizontal and vertical spacings $\Delta P'$ and $\Delta z'$ both smaller than 1. There is an optimum though: downwind distance should not be so large that methane plumes from the facility of interest are missed, and measurements very



far downwind are also unfavorable because of the lower methane concentrations expected, which will lead to a lower signal-to-noise ratio. It appears to us that values of $\Delta P'$ and $\Delta z'$ between 0.1 and 1.0 are optimal.

*Author contributions.* The ideation and management of the research project was done by RI and THM, who are the main authors of the manuscript. The simulation studies and all data analyses were done by THM, with help of MJ and BVDK, under guidance from RI. BS, KD and AC provided historical data from SeekOps and facilitated the data analysis; SC did the
same for historic data from Scientific Aviation. All authors were involved in reviewing the paper.

*Data availability.* All the codes and data used in this work will be made available online.

*Competing interests.* The corresponding author has declared that none of the authors has any competing interests.



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



## Appendix A: Sensor description

The main paper describes two historic data sets of methane emission measurements: one set from Scientific Aviation and one set from SeekOps. Both data sets contain concentration meauserements taken by a high-precision methane sensor mounted on a quadcopter drone. The present section presents background on the type of sensor that has been used in these deployments. The text below is applicable to the sensor developed and used by SeekOps; Scientific Aviation use a commercial off-the-shelf sensor that has similar characteristics.

The SeekIR sensor, used by SeekOps is the culmination of years of research and commercial development, initially at the United States National Aeronautics and Space Administration (NASA) Jet Propulsion Laboratory (JPL). The technology was originally developed for the Mars Curiosity Rover to look for evidence of microbial life and was thus developed as extremely sensitive to methane enhancements above background levels, (Webster , 2005). In 2017, the technology was spun-out of JPL and commercialized for the broader energy industry, including traditional oil and gas, biogas/landfill gas, and renewable natural gas (NASA , 2019). It was subsequently validated by blind controlled release tests performed at the Methane Emissions Technology Evaluation Center (METEC) in Colorado, where the sensor was described as the most successful in detecting and quantifying leaks, with no false positive and no false negatives (Ravikumar et al. , 2019).

The sensor operates on the principle of absorption spectroscopy, using a tunable diode laser (TDL) within an open cavity bounded by two mirrors that give a suitable path length for the laser to ensure high sensitivity to the absorption in the presence of methane molecules. This physical process is described by the "Beer-Lambert Law", which describes how the spectral intensity measured at a specific wavelength after passing through a sample can be used to characterize physical parameters based on an initial spectral intensity and absorption path length (Hanson et al. , 2016). When parameters such as pressure, temperature, wavelength, and path length are measured or known, the concentration of the species of interest can be calculated by this change in spectral intensity. In the system described, the initial spectral intensity source is a TDL and the sensor measuring the change in spectral intensity is a photo-voltaic detector sensitive at the same spectral region as the laser light source. The detector measures the remaining photons that were not absorbed by the methane molecule(s) as part of the absorption process. A spectroscopic approach, Wavelength Modulation Spectroscopy (WMS), is further employed by modulating the laser intensity and wavelength at a known frequency. WMS ensures the optimal signal-to-noise ratio (SNR) and a "calibration-free" method of laser diagnostics, allowing for calibration factors that persist for the life of the instrument that are determined at the time of manufacture. With appropriate filters for elevated frequencies, the detected response can be demodulated to determine the concentration of methane that has been sampled within the open cavity (Figure A1).

The SeekOps instrument used in this study additionally utilizes a multi-pass optical cell design, known as a Herriott Cell, that increases the laser path length, further increasing sensitivity of changes to concentration, temperature, or pressure, depicted in Figure A2. The Herriott Cell design is comprised of two concave mirrors of a high reflectivity



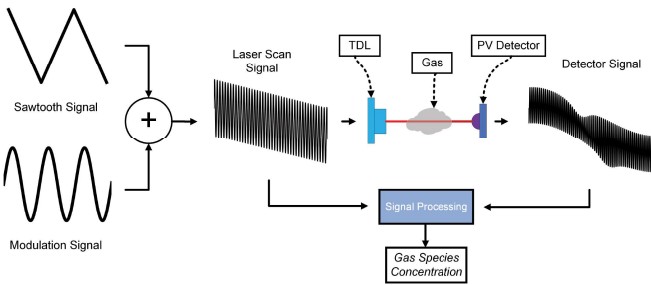

**Figure A1.** Simplified drawing of the implementation of Wavelength Modulation Spectroscopy (WMS) using a tunable diode laser (TDL) for gas sample diagnostics.

and matched to reflect the proper wavelength of light, allowing the beam to bounce multiple times inside the cavity helping to increase the SNR and further reduce measurement uncertainty.

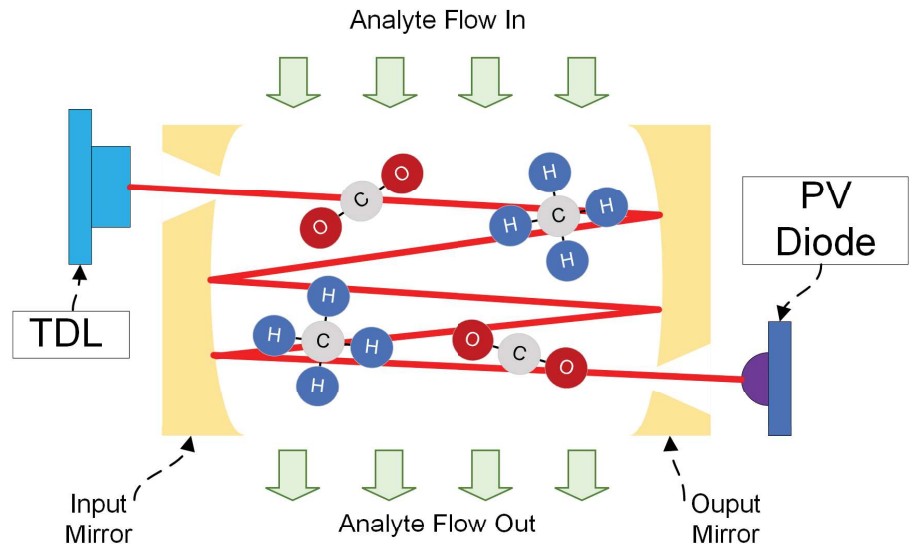

**Figure A2.** Drawing of the sensor operating principle of laser absorption spectrometry to determine methane concentration.

Finally, great efforts have been taken to make the optical sensor robust for operations in the oil and gas sector. To this end, the sensor is mounted in a strong housing that has undergone strenuous shock and vibration testing, indicative of its operational environment when deployed on a quadcopter drone and transported to and from a





facility of interest. Modern drones have multiple anti-collision sensors that can enable the sensor to be flown as close as necessary to the equipment without compromising any ongoing operations.

A picture of the SeekOps sensor is shown in Figure A3. The sensor is mounted on an approximately 1 m long extended steel branch attached to the drone, which makes the sensor easily accessible in case of maintenance requirements. Data acquisition takes place with the branch pointing into the wind, which allows the collection of pristine samples of air, uncontaminated by any prop wash (i.e., the airflow generated by the drones own propellers). Methane concentration data from the sensor are streamed and displayed in real-time to a ground control system (GCS). This

allows for on-demand viewing of plume methane enhancements by the drone pilot, so that dynamic adjustments to the survey geometry, like the horizontal and vertical spacing between concentration measurements, can be made if needed.

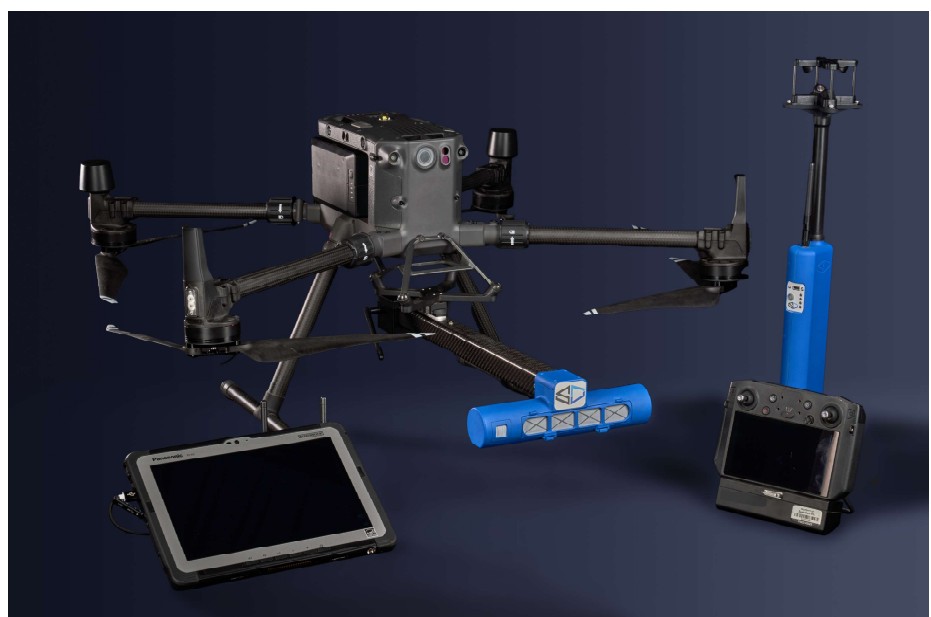

**Figure A3.** Methane sensor showing location of integrated GPS, LiDAR and attached to the quadcopter drone.

## Appendix B: Cylindrical flight pattern

In this section, the results from the simulation of a drone flying in a cylindrical pattern are presented, similar to

Section 3 for 2-D curtain flight path. We simulated the drone flight for different scenarios with the objective of assessing the importance of various sources of uncertainty as the following:





– the position of the emission source relative to the center of the cylindrical flight paths.

– the existence of multiple sources within the cylinder.

– horizontal and vertical spacing in the methane concentration measurements.

Again, it is assumed that the concentration field is stationary during the time of the measurements. This can relate to a constant Gaussian plume traveling horizontally, in a time-averaged horizontal wind field whose speed can vary as a function of the vertical coordinate.

**B1  Ideal case**

The ideal simulation case is considered as the following, similar to the ideal case considered for the 2-D curtain
flight, Section 3.1:

– Constant wind field with westerly wind with the speed of $5\,\mathrm{m/s}$.

– One circular flight path passing through the center of the plume.

– Equidistant drone measurements in the horizontal and vertical directions.

– No measurement error in simulated concentration, wind field and drone measurement locations.

The radius of the cylinder, opening angles and wind field are chosen such that they reflect a situation encountered in practice. The source with the emission rate $5\,\mathrm{kg/h}$ is considered. The cylindrical flight pattern is often used for offshore applications where the horizontal and vertical opening angles of the plume are smaller than those encountered in onshore applications and thus the opening angle of $2.5°$ is chosen as a realistic value for the 'Ideal case' (base case). Figure B1 shows the 3-dimensional concentration plot due to this source on a cylinder with a radius of $10\,\mathrm{m}$.
Note that this radius is much smaller than the $\approx 300\,\mathrm{m}$ that Flylogix/SeekOps normally uses in surveys of offshore platforms. This should not affect the main results, because the radius will appear as the length scale when considering the spacing. With a radius of $10\,\mathrm{m}$, it is reasonable to assume that the methane plume is stable, as the methane in the plume only takes a few seconds to go from source to sensor. At longer distances of $\approx 300\,\mathrm{m}$, the transit time is of the order of $\approx 1\,\mathrm{min}$ and the wind direction can vary considerably during that time.

**B2  The importance of turbulent diffusion on mass emission calculation**

For a cylindrical flight pattern, the true emission rate cannot be exactly retrieved. In Figure B1, there is an error of $0.006\,\mathrm{kg/h}$ (equal to a percentage of $0.12\%$) even for a very fine spacing in the horizontal and vertical directions. This error does not exist for the 2-D curtain situation as the true emission rate is retrieved from the mass balance equation. The emerged error term is due to the effect of diffusion, see the third term in the right-hand side of
Equation (4). If the flight path is cylindrical, the direction of the unit vector $\mathbf{n}$ varies along the flight path, and





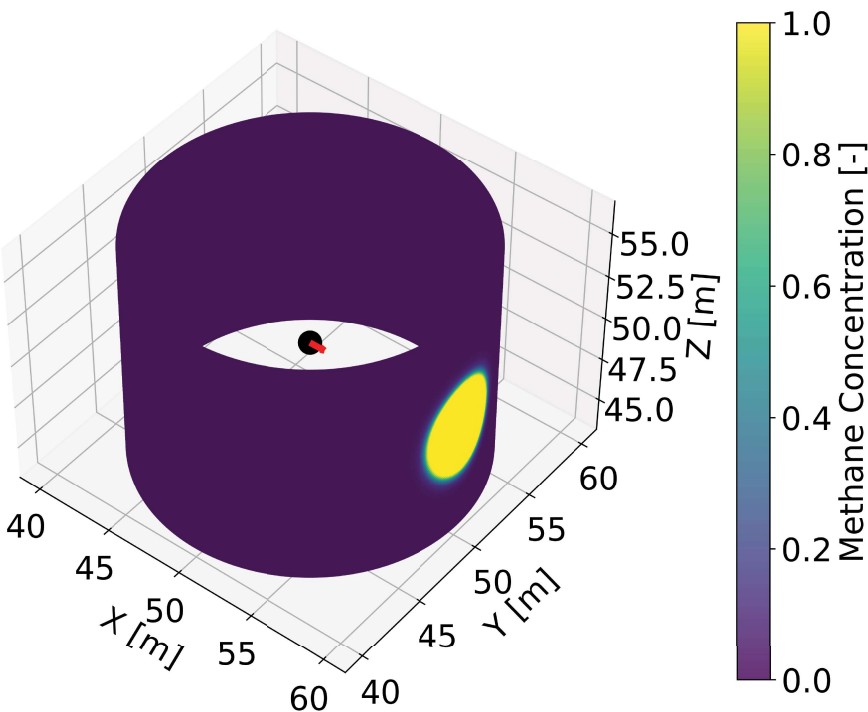

**Figure B1.** 3-dimensional normalized methane concentration above background on a cylinder with a radius of $10\,\mathrm{m}$ for a source located at the center of the cylinder (shown by black). The wind velocity is shown with a red vector. The Gaussian plume is assumed to have 5° horizontal and vertical opening angles. There is an angular spacing of 0.5° between the measurement points and a vertical spacing of $0.3\,\mathrm{m}$.

the contribution of $\mathbf{n}\cdot\boldsymbol{\nabla}c$ does not vanish along the integral. Since this term is not normally included in the mass balance equation (Equation (6)), this results in an underestimate as $\mathbf{n}\cdot\boldsymbol{\nabla}c$ is typically larger than 0 in the case where a source is located in the centre of the cylinder. This error cannot be eliminated, even in the limit of perfectly accurate vertical and horizontal measurements; to first order, the error is proportional to the diffusion coefficient $K$.

**B3 Position of the emission source relative to the center of the cylinder**

In Section B1, we assumed that the source is located at the centre of the cylinder, however, in practice this might not hold. It is thus beneficial to assess the impact of such assumption on the calculated emission rate. Shown in Figure B2 is the error percentage for varying shift of the source from the centre as a percentage of the radius of the cylinder. The wind direction is also shown with green vector. For the situations where the source moves towards





the cylinder edge in the upwind direction (i.e., towards the left in Figure B2), the error in the calculated emission rate decreases due to the decrease in the effect of diffusion. However, if the source gets very close to the cylinder on the right-hand side, the term $\frac{dc}{dy}$ increases (as the concentration increases). One should bear in mind that that the neglect of diffusion for the cylindrical flight pattern typically has a small contribution to the error compared to the other sources considered.

Figure B2 can be also used to calculate the total error in the calculated emission rate in case of having multiple sources within the cylinder. We will illustrate this with an example, assume two sources $A$ and $B$ with true emission rates of $\dot{m}_A$ and $\dot{m}_B$ respectively. The calculated emission rate error percentage for each source, referred to as $\epsilon_A$ and $\epsilon_B$, can be obtained from Figure B2 using their position relative to the centre of the cylinder as a percentage of the radius of the cylinder. Using these calculated emission rates, the total error in the situation where multiple

sources exist within the cylinder can be obtained as: $\epsilon_{A+B} = \frac{\epsilon_A \dot{m}_A + \epsilon_B \dot{m}_B}{\dot{m}_A + \dot{m}_B}$.

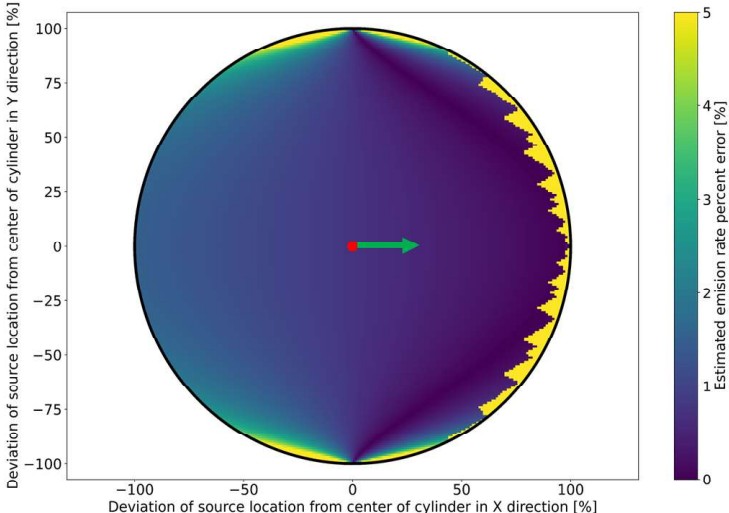

**Figure B2.** Calculated emission rate error percentage for shifted source in $X$ and $Y$ directions from the centre of cylinder as a percentage of the cylinder's radius. The wind direction is shown with the green vector and the centre of is indicated by red circle.

## B4   Horizontal and vertical spacings of cylindrically shaped drone measurements

### B4.1   Non-equidistant vertical spacing

Similar to 2-D curtain scenario, we assessed the effect of non-equidistant vertical spacing between the flight coordinates in the vertical direction, see Section 3.2 for the 2-D curtain. Similar to Section 3.2.1, $L$ random samples from a





Gaussian distribution with zero mean and a standard deviation $\sigma_l$ are drawn, see Equation (14). For each simulation
scenario (i.e., each standard deviation) 300 independent runs are considered. Figure B3 shows the box-whiskers for
the emission rate percentage error for non-equidistant vertical spacing. Similar to Figure 10, no bias emerges in the
emission rate estimates with increasing variation in the vertical spacing. However, individual measurements can have
errors up to almost 15% if the standard deviation in the vertical spacing is equal to the nominal vertical spacing.

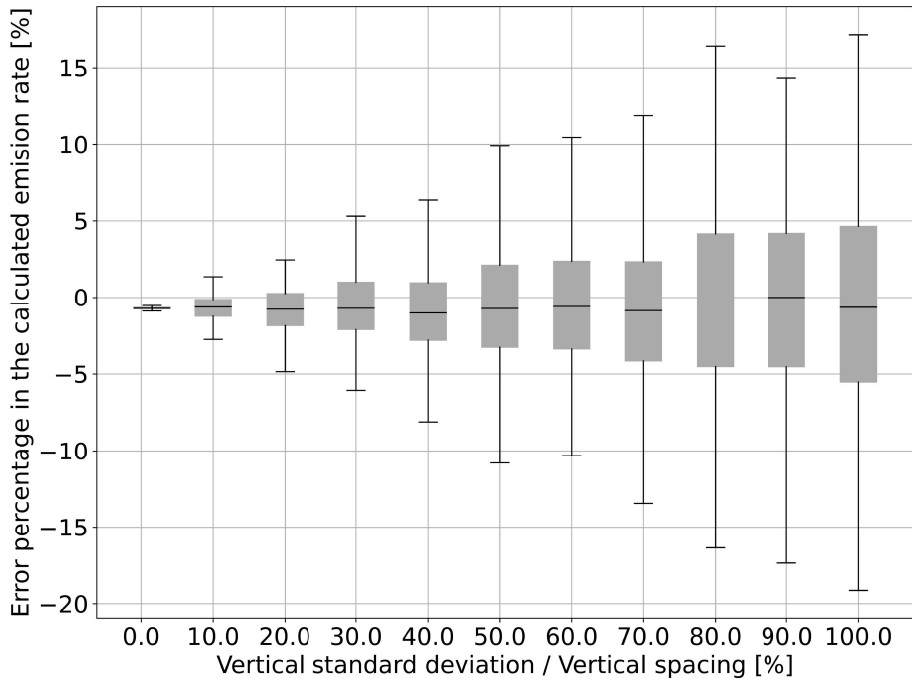

**Figure B3.** Box-whisker plot for the calculated emission rate percentage error for non-equidistant vertical spacing between
the flight coordinates, for the cylindrical flight paths.

**B4.2    Missing the plume center in relation to Horizontal and vertical spacings**

In Section B1, it is assumed that one circular flight path passes through the centre of the plume in the vertical
direction and that there exist concentration points on this circle corresponding to the maximum concentration of the
plume on the cylinder pattern. As discussed in Section 3.2.2, these assumptions are easily violated as the location
of the source is not exactly known. Therefore, here we will assess the potential uncertainty induced by missing
the maximum concentration of the plume for different horizontal and vertical spacing of the drone measurements.
The resulting error is a function of the horizontal and vertical spacings as the drone is more likely to miss the





highest concentration of a plume if the horizontal and vertical spacing is large. For the cylindrically shaped drone measurements the angular spacing $\Delta\theta$ which can be converted to the horizontal spacing in meter using the radius of the cylinder ($r$) as $r\Delta\theta$. To assess the effect of missing the plume, for each angular ($\Delta\theta$) and vertical ($\Delta z$) spacing of the drone measurements, 15 different shift values relative to the plume center (projected on the cylinder) ranging from $-\frac{\Delta\theta}{2}$ to $+\frac{\Delta\theta}{2}$ in the horizontal and from $-\frac{\Delta z}{2}$ to $+\frac{\Delta z}{2}$ in the vertical directions are considered. For each shifted location, the emission rate error is calculated, i.e., 225 (15 $\times$15) instances of calculated emission rate errors. For all the locations except the one where one flight line passes the centre of the plume, the maximum concentration is missed. Therefore, the emission rate is always underestimated (except for one case) and the maximum error representing the worst situation is considered for each horizontal and vertical spacings. Figure B4 shows the maximum error percentages for a cylindrical flight pattern for the horizontal and vertical opening angles of the Gaussian plume equaling 2.5° (corresponding to a realistic scenario offshore).

Similar to the generic case presented for 2-D curtain, Figure 4, the vertical spacing is presented as dimensionless parameter $\Delta z'$. For the horizontal spacing, we use the angular spacing in degrees which makes it invariant of the radius of the cylinder. Increasing both the vertical and horizontal spacings between the coordinates lead to an increase in the calculated emission rate error. For the wider plume, the errors are smaller as more concentration points exist within the plume for calculating mass balance equation.

### B5 Time-variations in wind speed and/or direction

In Section B1, it is assumed that the wind velocity field is time-invariant, however, in practice this assumption is rarely true, i.e., instantaneous changed in the wind velocity can occur. Similar to Section 3.4, random samples are drawn from Ornstein-Uhlenbeck process with a pre-defined standard deviation for the wind speed and the wind direction. For each standard deviation, 300 independent runs are considered and the corresponding box-whisker plot for increasing standard deviation are shown in Figure B5. The results are very similar, both qualitatively and quantitatively, to the results obtained for the 2-D curtain pattern (Figure 6).

### B6 Other potential sources of error

There are other potential sources of error and uncertainty occurring in measurements from drones flying in a cylindrical flight pattern. Many of these sources of uncertainty will be the same as for the 2-D curtain pattern. One notable source of uncertainty that is specific to cylindrical flight patterns is the long time that it takes to take relevant concentration measurements. In comparison with 2-D patterns, the cylindrical flight pattern requires a relatively long flight-time to come back to regions where concentration peaks are located. There is therefore an increased likelihood that wind directions will vary during these measurements, resulting in more uncertainty in plume locations and the risk to miss concentration plumes, or to measure the same plume during multiple transects.

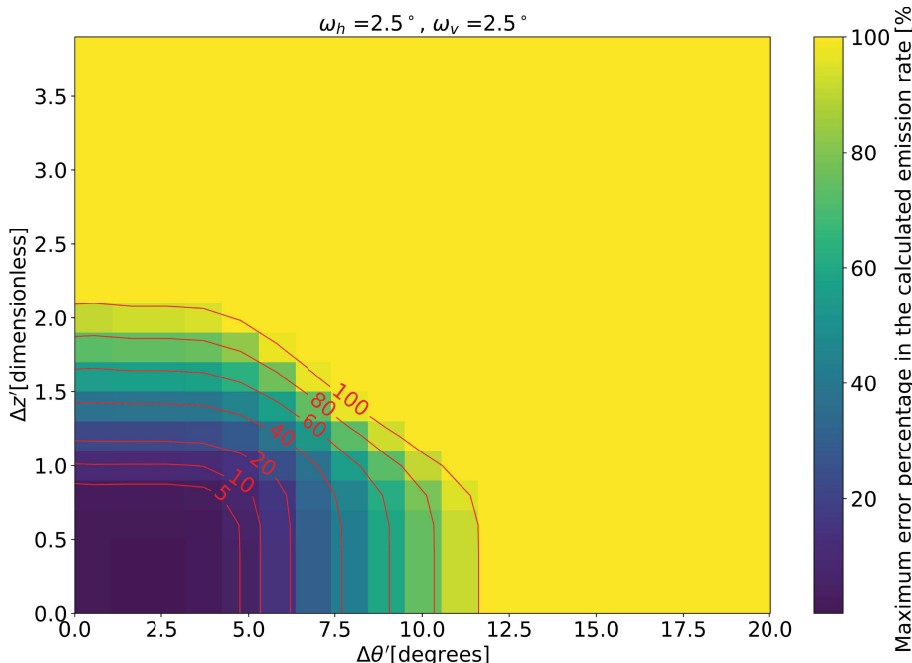

**Figure B4.** Maximum calculated emission estimate error for varying factors of horizontal and vertical spacings expressed as degrees and dimensionless parameter for a plume with vertical and horizontal turbulence of 2.5°. For each horizontal and vertical spacing, maximum value over 225 calculations corresponding to different locations of the flight lines relative to the source is shown. Shown with red are the error contours.

**Appendix C: Wind data calculation**

In order to allow the uncertainty analysis as explained in Section **Simulation results** for 2-D curtain flights, the raw wind data needs to be converted into an absolute wind speed and its root-mean-square fluctuation, and a plume opening angle. The conversion process is graphically illustrated in Figure C1. The absolute wind speed $|u|$ is determined from the perpendicular components of the average wind vector, $U$ and $V$ as follows: $|u| = \sqrt{U^2 + V^2}$. The average wind direction from East (in radians), is then computed from: $\theta = \tan^{-1}\frac{U}{V}$. The root-mean-square fluctuations in the direction of the plume, $\psi'$ in m/s, can be computed from the fluctuations in the East-West and North-South directions respectively denoted by $u'$ and $v'$: $\psi' = u'\cos\theta + v'\sin\theta$. The dimensionless fluctuations divided by the wind speed are given by: $\frac{\psi'}{|u|}$. The root-mean-square fluctuations perpendicular to the plume, $\Phi'$ (also in m/s), is then given by: $\Phi' = -u'\sin\theta + v'\cos\theta$. Finally, the approximate opening angle of the plume in radians is given by: $\tan^{-1}\frac{\Phi'}{|u|}$.





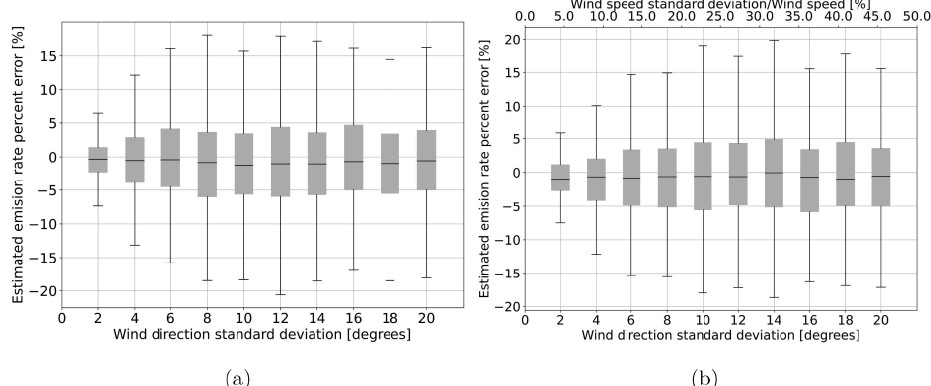

(a)                                                            (b)

**Figure B5.** Box-whisker plot for the calculated emission rate error for increasing standard deviation of a) time-varying wind direction and b) time varying wind velocity (both direction and speed at the same time). Cylindrical flight pattern with a radius of 10 m for a source located at the centre of the cylinder is considered.

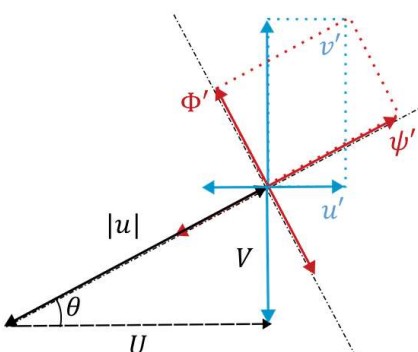

**Figure C1.** Graphical illustration of the conversion of perpendicular wind data (average wind vectors $U$ and $V$, with root-mean-square. fluctuations $u'$ and $v'$) into an average absolute wind speed $|u|$, root-mean-square. fluctuation in the direction of the plume $\Phi'$ and the plume opening angle: $\tan^{-1}\frac{\Phi'}{|u|}$.



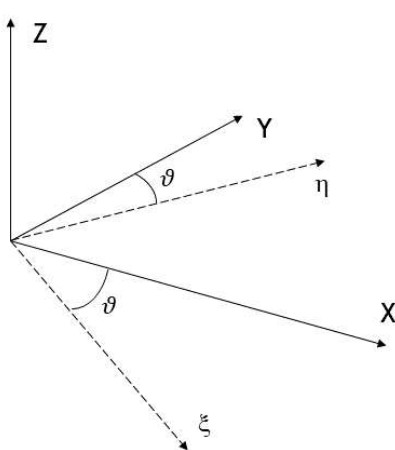

**Figure C2.** Graphical illustration of the Local Geodetic coordinate system ($XYZ$) and wind direction aligned coordinates system ($\xi\eta z$).