# Peer review of "Quantitative estimate of sources of uncertainty in drone-based methane emission measurements"

_EGUsphere, 2024_

## Author Comment (AC2)

**Cover letter to the Editor**

Dear Editor,

Thank you very much for collecting the responses from the two reviewers on the original version of our manuscript egusphere-2024-1175. We have attempted to address all their comments; please see the replies to the individual reviewers below. We have prepared a revised manuscript, which we are planning to submit in AMT's online portal together with this letter.

We believe that our paper has improved significantly because of the reviewers' suggestions, and we would like to thank you again for facilitating the review process.

Reviewer #2 (Steven van Heuven) raised a question about competing interests; see below. We have answered his question accurately and transparently below. We kindly request you to also look at his question and our answer; we appreciate any suggestions from your side on whether the "Competing interests" section is the right place to describe the collaboration between the various parties that resulted in our manuscript.

We look forward to hearing you about the next steps in the process.

Kind regards, on behalf of all the authors,

Rutger IJzermans

**Reply to Reviewer #1, Joseph Pitt**

Dear Dr. Pitt,

Many thanks for your thorough review of our manuscript. We have attempted to address all your comments; please see our replies in blue below. The revised manuscript has been attached to this letter, too. We believe that our paper has improved significantly because of your suggestions, and we would like to thank you again for taking the time and effort to review it.

**Comments from Reviewer 1, Joseph Pitt:**

This study provides a thorough assessment of the sampling errors associated with facility-level mass balance emission estimates made using drones. This approach to estimating emissions is becoming increasingly popular due to a combination of improved lightweight measurement technology and an increased focus on facility-level reporting by site operators (e.g. through OGMP2.0). It is therefore important to understand the various sources of uncertainty associated with these estimatsaes so as to better design future mass balance experiments and to interpret the results. This manuscript represents a valuable contribution to this effort and offers useful practical guidance for future studies. The manuscript is very well written and clearly presented. I suggest that it should be published in AMT once the following minor issues have been addressed.

- The only substantial addition to the paper that I think is necessary is a discussion on how the background concentration was determined for the real-life examples. There are many ways that this can be determined, either statistically (e.g. some percentile) or spatially (e.g. from grid cells at the end of the plane), and it can be determined separately for individual transects/curtains or a single background can be used for the whole flight. Maybe the enhancements are large enough that different choices of background don't matter in these cases, but even if that is the case it is important to explain how the background was determined. I think it would also be useful to demonstrate the impact of some different plausible background calculation methods, even if only to show that it doesn't matter here.

  Thanks for your comment. Indeed, that was a point that required more discussion than we initially gave it in the first version of the manuscript.

  For the data from Scientific Aviation (discussed in section 4.1), the background was calculated using 10% percentile of the concentration measurements. For the case with basic postprocessing, all the concentration measurements were used in the calculation of the background. For the situation where we divided the flight into multiple curtains, the background was calculated separately for each curtain, now using the 10% percentile of the concentration measurement within the curtain. This point is now clarified, and the calculated background is added to Table 1.

  We also added the following text to the Section 2.3 ("Drone-based methane measurements") as:

*The atmospheric background can be calculated statistically (e.g., some low percentile of the concentration measurements), McKain et al. (2015 and Plant et al. (2022), or spatially (e.g. from grid cells at the edges of the plane), Mays et al. (2009) and Conley et al. (2017). In this paper use is made of the former approach.*

- L18 – typo "a course spacings"
  Corrected.

- L37-39 – I can't quite work out how these values relate to table 3 in Saunois et al. The 60% anthropogenic value is presumably from the top-down estimates, but the 13% value for oil and gas must be derived from the bottom-up values? It would be good to add some more context, and also to emphasize the large uncertainties associated with these values.

  Thank you for the comment. We agree with you that our paper was not clear on the origin of the numbers quoted, so we have decided to provide a slightly longer explanation as follows:

  *According to bottom-up estimates over the period 2008-2017 by (Saunois et al. , 2020), about half of total methane emission sources are anthropogenic (366 Tg/y out of 737 Tg/y), predominantly from agriculture and waste (206 Tg/y); oil and gas production accounts for 22% of total anthropogenic emissions (80 Tg/y). Although all these numbers are subject to high uncertainty, the estimates do show that reducing methane emissions from the oil & gas industry can have a significant effect on limiting climate change in the coming decades.*

- L43 – check sentence grammar
  Corrected.

- L66 – remove second "variations"
  Corrected.

- L224-225 – remove "correspond" or "relate"
  Corrected.

- L319 – why do large variations in wind speed always lead to an underestimate? This isn't immediately obvious to me so it would be good to expand on it here.

  Thanks for your comment. This point probably required more explanation than we provided in the first version of our manuscript. We have now expanded the description of the impact of time-varying wind direction and wind speed in section 3.4 ("Time variations in wind speed and/or wind direction").

- L429 – should this be "blue to yellow"?

  Thanks for your comment; you are right. It has been corrected in the revised version of the manuscript.

- L431-432 – should this be "orange circles"?

  Thanks for your comment; you are right. It has been corrected in the revised version of the manuscript.

- Figures 11 and 12 – references to these figures in the text on page 23 are mixed up in multiple places.

  Thanks for your comment; you are right. It has been corrected in the revised version of the manuscript.

- Also in Figure 12a, why do all the simulations have an error of 3.25%?

  Thank you for the question. The result shown in Figure 12a is related to the similar outcome shown in Figure 6a, which showed that changes in wind speed lead to significant errors only when the wind speed standard deviation of larger than 40%. When generating Figure 12a, we only used data points from flights in which the wind variation was low; this typically holds for situation where the wind is steady. The relative variations in wind speed for this subset of flights were found to be limited to the range 0-25%. As a result, the error in the estimated error because of wind speed variations was found to be close to 0% too.

  In the first version of the paper, the 3.25% was actually a typo as it should have been $3.25 \times 10^{-5}$% (i.e., five orders of magnitude smaller); the "$\times 10^{-5}$" was not included in the graph. We have updated the graph in the revised version of the manuscript.

- Figure B2 – the discussion of this figure confuses me. The text says that the error decreases towards the left of the figure, but that does not appear to be the case.

  Thanks for your comment. The text has been now modified.

- A few figures (e.g. Fig. 4, Fig5) have slightly clipped legends/axes labels and would look neater if replotted.

  Thank you for the suggestion. This has been done now in the revised version of the manuscript.

**Reply to Reviewer #2, Steven van Heuven**

Dear Dr. Van Heuven,

Many thanks for your thorough review of our manuscript. We have attempted to address all your comments; please see our replies in blue below. The revised manuscript has been attached to this letter, too. We believe that our paper has improved significantly because of your suggestions, and we would like to thank you again for taking the time and effort to review it.

**Comments from Reviewer 2, Steven van Heuven:**

This manuscript purports to present a rigorous characterization of the effects of various sources of error on the estimates of rates of point source emission of trace gases, ostensibly in order to assign post-hoc uncertainties to such estimates. Undoubtedly these uncertainties are currently not well established and direly needed, so an attempt to provide an analytical framework is laudable and much appreciated.

Having said that, I find it hard to assess how much of a /general/ framework is offered here. The derived uncertainties appear to pertain – at least in part – to the measurement, simulation (in part idealized) and processing methodologies that are particular to this study. I believe the manuscript may benefit from more clearly delineating between theory and practice (some suggestions below), and striking a more modest tone where it comes to applicability of its finding. Certainly, the opening wording of the discussion ("[the presented framework] allows to back-calculate errors and uncertainties in historic data") reads a little presumptuous. Other sections, on the other hand (for instance the abstract) are more balanced.

Thank you for your comments. We have rephrased the first paragraph of the Discussion section as follows:

Earlier text:

*The theoretical framework presented above allows for a post-hoc analysis of historic data, which allows the evaluation of the quality of the measurement from a methane emission quantification survey. Potential errors and uncertainties in the measurements can be identified. With this, it is possible to back-calculate errors and uncertainty in the results from historic measurement data.*

New text:

*The theoretical framework presented above allows for a post-hoc analysis of methane emission quantification surveys. With this, a number of potential errors and uncertainties in the measurements can be identified.*

The writing style of the manuscript is excellent. More importantly, the manuscript features considerable rigor in physics and math and some interesting normalization and scaling ideas. These may well aid readers in their own assessment of mass fluxes from UAV measurements. I therefore recommend publication in AMT, with the request that the following points are addressed.

- I balk a bit at the "the authors have no competing interests"-statement; all authors work for commercial vendors or beneficiaries of the technology that through this publication obtains increased credibility. If that's not a competing interest I do not know what is...

Thank you for your comment. Before submitting the first version of the manuscript, we had carefully considered our declaration of not having competing interests. We are happy to share our considerations for that.

Each of the authors is employed only by the company mentioned in their respective affiliation. The content of the paper is obviously aligned with the business interests of the companies mentioned, but we do not think that there is any issue of competing interests here. This would be the case only if individuals or companies had separate interests that would not be clear from the affiliations.

Let us briefly explain the background as to why we, as authors, started to write a paper together. Over the last couple of years, several UAV surveys were performed at Shell assets by SeekOps and by Scientific Aviation (now part of ChampionX), and these surveys were typically paid for by Shell or its affiliates. The analysis of the results from these surveys showed that there were common sources of potential errors. The authors recognized that there was a shared interest to better understand the uncertainties associated with UAV-based methane emission surveys, and this has resulted in the present paper. We hope that the findings of our paper will not only benefit the companies in the affiliation list, but also the energy industry at large.

One further clarification: GTI was not involved in this paper. Abigail Corbett is included in the authors' list because of her contributions from the time when she was still employed by SeekOps. Since she only started to work for GTI recently, we have made that clear in her affiliation in the revised manuscript.

No money was exchanged between any of the parties in the writing of the current paper.

In summary, we appreciate and understand your question, but we still believe that the original statement was accurate. We have brought this topic to the attention of the AMT editor as well, to seek further guidance and clarification.

- As frankly admitted by authors, several major sources of uncertainty are not covered in this paper. Obviously, that cannot be expected of any single study. Nonetheless, consider reflecting that in the title, which currently reads rather all-encompassing. E.g., use "*several* sources of unc.".

  Thanks for your comment. We have changed the title accordingly.

- L86. Not that it matters much, but do you mean "z=0 at the emitter" or "z=0 at ground level" (i.e., same as z=Z)? If you mean the latter, consider writing the latter. "z = 0 at ground level" is clear, while "z=H where H is source height" is (to me) confusing.

  Thanks for your comment; this has been made clear in the text now.

- In section 3.2.2. I suspect that the choice of nearest-neightbor interpolation for filling the grid cells of your curtain may make your method susceptible to bias from 'missing the plume' in a way that using other interpolation methods (i.e., kriging) would not. As a user I would be very much helped by an assessment of, say, kriging-vs-nn_interp. A rigorous treatment of that may be beyond the

scope of this manuscript, but a qualitative treatment of the pros and cons of nn_interp compared to objective mapping, and why you chose nn_interp, may be warranted here. Also, please provide visual example(s) of the resulting interpolation of a flight trajectory, ideally one for "crossing" and one for "missing" the plume.

Thanks for your comment. It is a good suggestion to look at the effect of the choice of the interpolation method. In the simulated cases, however, the choice of the interpolation method does not play a significant role, because the distribution of the measurement is ideal there. Still, for the completeness, we modified Section 2.4. as follows:

*The measured wind field is interpolated onto this grid using nearest neighbor method, referred to as $u_{j,k}$ (Sibson , 1981). Interpolating the concentration on the plane ($c_{j,k}$) can be achieved using different approaches depending on the spatial distribution of the drone measurements. For drone measurement with equidistant spacings in horizontal and vertical directions and lying on a plane, the nearest neighbor interpolation method can be used. However, for non-equidistant distributed drone measurements in a semi-random flight pattern, a more advanced interpolation technique is preferred, as the nearest neighbor might give a concentration that is too large or too small depending on the location of point in query and the drone measurement, as an example a Gaussian smoother or Gaussian process, (Price, 2011) and Rasmussen and Williams (2006). The emission rate is then obtained by substituting these interpolated values in Equation (6). It should be highlighted that if the plume is missed, the emission rate will be underestimated irrespective of the interpolation technique.*

In the real-life example using Scientific Aviation data, Section 4.1, the reason for choosing the Nearest Neighbor method is that this was the approach used by Scientific Aviation originally when calculating the emission rates. The text is now modified as the following:

*We use a Nearest Neighbor algorithm with the Euclidean distance metric to project the concentration measurements onto a vertical plane. This is only one choice for the data interpolation; other distance metrics such as Chebyshev and Minkowski distances can also be used in the Nearest Neighbor algorithm, and also other interpolation techniques such as a Gaussian smoother can be used as an alternative to the Nearest Neighbor process. Since Scientific Aviation uses the Nearest Neighbor for the projection of the concentration measurements on the vertical curtain plane in all their data analysis, we use the same interpolation technique to generate the results in this section. Section D shows that the Gaussian Smoother as the interpolation technique for this dataset results in similar emission rates.*

As you suggested, we assessed the impact of the interpolation method on the results. The comparison was made for measurement "A" where the calculation of the emission rate is presented per curtain, in Table 2, and individual curtains are discussed in detail in Figure 8. The maximum difference between the emission rates estimated for the four curtains using the two interpolation methods is 8.40 g/h and the impact of the average emission rate is 0.44 g/h (1.4% of the total emission rate of 30.8 g/h). Our tentative conclusion from this analysis is that the choice of the interpolation method has a relatively small impact on the mass emission estimate. The results of the comparison have now become the new "Section D" in Appendix.

- In interpreting section 3.4, this reader misses detail on how the wind field variability-modeling is set up, and there's no specific reference to get started with. Section 3.4 sort of suggests that during each of 300 models runs, continuously emitted methane is carried from the source by a time-varying wind field. That may produce realistically varying, patchy concentrations at the curtain /within/ each simulated flight, and that would be great (and hard to work with). But from the methods-section, I get the idea that you perform 300 runs /between/ which the wind speeds differ and the idealized gaussian plume is located at slightly different locations every run. Such data would be much easier to work with but may present an underestimate of the effect of wind field variability. Consider expanding and/or illustrating what a typical simulated curtain looks like, to aid the reader in interpretation of your results.

  Thank you for this comment. It looks like your first interpretation (citing "section 3.4", with the time-varying wind field that occurs in the course of a single simulation) is correct, but we agree that there is no harm in better explaining what we have done in the simulations.

  We have therefore expanded the description a little, also referring to the Methodology section in section 2.4 where the simulation setup was first presented.

  Old text:

  *To take this issue into account, we consider a 5 m/s westerly wind at the start of the flight and randomly draw samples from an Ornstein-Uhlenbeck process with a defined standard deviation in the wind speed and/or the wind direction to simulate a time-varying wind field. For each standard deviation, 300 independent runs are considered, and the results are shown for time-varying wind direction (a), wind speed (b) and their combination (c) in Figure 6.*

  New text:

  *To take this issue into account, we consider a large number of simulations in which a Gaussian plume changes randomly according to the Ornstein-Uhlenbeck process described in section 2.4. The factor $\tau$ is chosen equal to 30 s in all simulations. Each simulation starts with a westerly wind of 5 m/s, but it gradually evolves into a fluctuating wind field with a statistical standard deviation in the wind speed and/or in the wind direction in accordance with a pre-defined factor $\sigma_\chi$ (see Eq. (13)). For each standard deviation, 300 independent simulation runs are performed, and the results are shown for time-varying wind direction (a), time-varying wind speed (b) and their combination (c) in Figure 6.*

- On that note: there's a bit of a surprising scale difference between the modeling setup (5 kg/h, 5 m distance) and the purported use in UAV work, which would typically necessitate larger distances. In the modeling setup, there is (as I understand it) a ~1 second travel time from source to curtain, which would not allow for patchiness to develop. Only a sort-of-random-walking circle could exist, whether within or between runs - correct? (ah, all this is mentioned later on nevermind).
- I believe that in particular this treatment of wind variability does not translate very well to practice, and that should be noted more clearly right away, not only later on in 3.5 ("other potential sources").

Thanks for your comment. A detailed explanation is now given in the text for the effect of wind direction variability, wind speed variability and both.

- L319 Please provide a rationale for why the emissions estimate goes down with wind field var, similar to the (satisfactory!) explanation for the 'curtain angle effect' in fig. 5.

Thanks for your suggestion. As mentioned above, a new explanation is now given in the text for the effect of wind direction variability, wind speed variability and both.

- There is no mention here of the background concentration – was that provided by ScAv? If not, how was that obtained, and what uncertainty is associated with the choice of background? Some later on for the SeekObs flights.

Thanks for your comment. This point probably required more explanation than we provided in the first version of our manuscript.

For the data from Scientific Aviation, the background was calculated using 10% percentile of the concentration measurements. For the case without any postprocessing, all the concentration measurements were used in the calculation of the background. For the situation with the postprocessing and dividing the flight into multiple curtains, the background was calculated separately for each curtain using 10% percentile of the concentration measurement with the curtain. This point is now clarified, and the calculated background is added to Table 1.

We also added the following text to the Section "Drone-based methane measurements" as:

*The atmospheric background, $c_0$, can be calculated statistically (e.g., some low percentile of the concentration measurements), McKain et al. (2015) and Plant et al. (2022), or spatially (e.g. from grid cells at the edges of the plane), Mays et al. (2009) and Conley et al. (2017). In this paper use is made of the former approach.*

Regarding your comment on the uncertainty associated with the choice of background, for the flight "A" which have been discussed in detail in the manuscript the standard deviation of the background between the curtains is about 5 ppb. We also looked at all other flights from Scientific Aviation and the maximum standard deviation of the background between the flights was 9ppb. All this can be considered relatively insignificant in comparison with the maximum concentration enhancement of at least 500 ppb. Nevertheless, it was good that we addressed this potential source of uncertainty, so we would like to thank you for your comment; we have updated the text in section 4.1 of the revised manuscript accordingly.

- Figures 7a & 8abcd. Please indicate the wind direction (or, why not switch to the "ENZ"-system, as suggested L83-87?). For 8abcd, consider using identical perspectives - now they're all rotated, and it's hard to know how they compare. Same for color scale. Consider projecting the 'shadows' of the points onto the side and bottom walls of the domain, so reader gets a better sense of the 3D-structure of the point cloud/curtain. Is the source located at x,y,z = 0,0,0? Figure C2 and L86

certainly seem to suggest so. If so, was a sensor-carrying (i.e., big) UAV really flown just a few meters downwind of a ground-level source? Were these curtains not flown perpendicular to the wind? I have many questions about these figures. Please improve to clarify.

Thanks for your comment. These are good suggestions.

The wind direction is now indicated with an arrow in Figures 7 and 8. The coordinates where indeed in ENZ coordinate system and instead of x, y, z label, we should have used E, N and Z. This is now corrected in the revised version of the paper.

As for Figure 8abcd, instead of presenting them with different rotation angle, we now present all of them with the same rotation angle as Figure 7. We also projected the shadows on the side walls to give the reader a better sense of the three-dimensional structure, as you suggested. Regarding the color scale: since the emission rate and the background for each curtain were calculated separately, we decided to not change them to a common color scale as it would lead to a loss of information of the curtains with the lowest enhancements.

Regarding your question on the location of the source: no, the emission source was not located at (x, y, z) = (0, 0, 0). Although it was the controlled release experiment and we had the information on the location of the source, we treated the experiment as a blind trial when we did the inversion; we did not use (or, in fact: need) the information on the source location to be able to calculate a mass emission rate using the mass balance method.
From the drone measurement we obtained the latitude, longitude and altitude for each location and we transferred it to ENZ considering the minimum of curvilinear coordinates as the reference point for the transformation; the point (x, y, z) = (0, 0, 0) has no physical meaning.

We hope that the above clarification answers your questions. Thanks again for your comment; your suggestions led to a higher clarity of the figures.

- L357. "Curtains have been chosen such that they contain elevated concentrations and do not contain conflicting concentration values" (and then they match markedly well with the known rate). Judging from figure 9, you've discarded the latter half the dataset for that flight, which clearly contains elevated concentrations. The basis for discarding is thus the "confliction" criterion, which is not further detailed. Please do, to avoid being criticized for cherry-picking. Also, I think this criterion should be very self-explanatory, or to be shown to work well on the other datasets. If this is ad-hoc selection of curtains that work well, that certainly deserves mention (!).
  Thanks for your comment. We think that this relates to figure 7, not figure 9. We agree with you that the text can be improved here; the sentence you quote left a lot of room for interpretation, and we have now changed the description of our emission rate calculation.

  We chose the upward and downward curtains such that they fully cover the plume. This means that three curtains at the second half of the data were excluded as they do not fully cover the plume; although they show some higher concentrations than the background, they only partially covered the plume and the highest concentrations are seen at the edges of these curtains.

Including them in the estimate would result in an underestimation in the emission rate. The same criterion has been applied on all the data from Scientific Aviation and we believe this postprocessing approach is a more robust alternative to using all the concentration measurements without any postprocessing. This point is now clarified in the text (lines 401-405).

- Table 2: please indicate what % of the data was discarded and the basis for that.

  Based on the exclusion criterion discussed above 40% of the data point were excluded. This is now added to the text (line 403).

- L361 "overall average" - delete "overall"?

  Thanks for your comment. The text has been now modified.

- In general, I'm missing a description of how well the framework you developed applied to the 1001 flight of SeekOps. Were you able to find flight level spacings, curtains and the like for all 1001 flights, or was a certain percentage of the flights simply unprocessable? And was that all manual labour, or did you develop code to plow through the flights autonomously? If manual labour, how much "expert judgement" was needed and of what type? Just outlier-removal or more involved things like the "curtain selection" of 4.1?

  Thank you for the question. From the flight information and the wind measurements during each of the 1001 flights, we determined the overarching parameters that we used in estimating the various errors mentioned in the paper.

  We did not reprocess all the raw data from SeekOps to calculate mass emission rates again using the mass balance method, because such an exercise would have been of limited value since no information was available on the true emission rates of methane during these surveys. As all the 1001 measurements were done at actual industrial facilities, the true methane emission rates were not known independently. It would of course have been great if such "true" data had been available, but unfortunately this was not the case.

- Figure 9: Consider adding the words "Contours indicate ... etc. " to the start of the first line of the caption.

  Thanks for your comment. The figure caption has been now modified.
- Figure B1 (nitpicking): if the plume maximum value is normalized (see caption) to, supposedly, "1", why is it all yellow and no detail on the inside? The color suggests a much higher peak value in the middle of the plume. Consider adjusting or elaborating.

  Thanks for your comment. The figure is modified in the manuscript.

- Nice discussion and conclusions. Consider discussing how the main finding (strong dependence on vertical/horizontal spacing) might change with use of another mapping technique. Do you think you conclusion is 'universal'? (cf. L364-366).

  Thank you for the question. Your earlier question about the mapping technique led us to look further into the sensitivity of the methodology. Our in-depth analysis of measurement "A", described in section 4.1 and Appendix Section D, shows that the choice of the interpolation method has relatively small impact on the mass emission estimate.

  Although this may be seen as just anecdotal evidence related to one single measurement, there is a grain of 'universal' truth here, in the sense that some sources of error cannot be compensated for by different interpolations. For example, if the highest concentration enhancements are not measured because of a large vertical spacing between flightlines, then it will not be possible for any interpolation scheme to magically cause the high concentrations to re-appear in the data. The resulting estimate for the mass emission rate will underestimate the true mass emission rate regardless of the mapping technique employed.

  To make this point clear in the paper, we have included this in our discussion in Section D of the Appendix:

  *Our tentative conclusion from the analysis presented in this section is that the choice of the interpolation method has a relatively small impact on the mass emission estimate. Moreover, some sources of error cannot be compensated for by a different choice for the interpolation method: if the highest concentration enhancements are not measured because of a large vertical spacing between flightlines, then the resulting estimate for the mass emission rate will underestimate the true mass emission rate – regardless of the mapping technique employed.*

- Note that the "there is an optimum though"-sentences are present in almost (?) identical wording on both sections.
  Thanks for your comment; well spotted.

  The text in the Discussion section has been modified as:

  Old text:

  *There is an optimum though: downwind distance should not be so large that methane plumes from the facility of interest are missed, and measurements very far downwind are also unfavorable because of the lower methane concentrations expected, which will lead to a lower signal-to-noise ratio. It appears to us that values of $\Delta P'$ and $\Delta z'$ between 0.1 and 1.0 are optimal.*

  New text:

  *On the other hand, however, downwind distance should not be so large that methane plumes from the facility of interest are missed. Measurements very far downwind are also unfavorable because*

*of the lower methane concentrations expected, which will lead to a lower signal-to-noise ratio. We recommend to use values of $\Delta P'$ and $\Delta z'$ between 0.1 and 1.0 in practical applications.*

We also changed the same text in the conclusion to:

*There is an optimum though: downwind distance should not be so large that methane plumes from the facility of interest are missed, and measurements very far downwind are also unfavorable because of the lower signal-to-noise ratio. It appears to us that values of $\Delta P'$ and $\Delta z'$ between 0.1 and 1.0 are optimal.*